# Tag-seq: a convenient and scalable method for genome-wide specificity assessment of CRISPR/Cas nucleases

Hongxin Huang [1,5], Yongfei Hu [1,2,5], Guanjie Huang[3], Shufeng Ma[3], Jianqi Feng [3], Dong Wang [1,2], Ying Lin [3✉], Jiajian Zhou [1✉] & Zhili Rong [1,3,4✉]

Genome-wide identification of DNA double-strand breaks (DSBs) induced by CRISPR-associated protein (Cas) systems is vital for profiling the off-target events of Cas nucleases. However, current methods for off-target discovery are tedious and costly, restricting their widespread applications. Here we present an easy alternative method for CRISPR off-target detection by tracing the integrated oligonucleotide Tag using next-generation-sequencing (CRISPR-Tag-seq, or Tag-seq). Tag-seq enables rapid and convenient profiling of nuclease-induced DSBs by incorporating the optimized double-stranded oligodeoxynucleotide sequence (termed Tag), adapters, and PCR primers. Moreover, we employ a one-step procedure for library preparation in Tag-seq, which can be applied in the routine workflow of a molecular biology laboratory. We further show that Tag-seq successfully determines the cleavage specificity of SpCas9 variants and Cas12a/Cpf1 in a large-scale manner, and discover the integration sites of exogenous genes introduced by the Sleeping Beauty transposon. Our results demonstrate that Tag-seq is an efficient and scalable approach to genome-wide identification of Cas-nuclease-induced off-targets.

[1] Dermatology Hospital, Southern Medical University, Guangzhou, China. [2] Department of Bioinformatics, School of Basic Medical Sciences, Southern Medical University, Guangzhou, China. [3] Cancer Research Institute, School of Basic Medical Sciences, State Key Laboratory of Organ Failure Research, National Clinical Research Center of Kidney Disease, Key Laboratory of Organ Failure Research (Ministry of Education), Southern Medical University, Guangzhou, China. [4] Bioland Laboratory (Guangzhou Regenerative Medicine and Health Guangdong Laboratory), Guangzhou, China. [5] These authors contributed equally: Hongxin Huang, Yongfei Hu. ✉email: linying0216@smu.edu.cn; zhoujj2013@smu.edu.cn; rongzhili@smu.edu.cn

The Clustered Regularly Interspaced Short Palindromic Repeats (CRISPR)/CRISPR-associated protein (Cas) system is a promising genome-editing tool that has been widely used in fundamental research and translational medicine[1]. However, the off-target cleavages of Cas nucleases are routinely observed and remain an obstacle for clinical applications, and several strategies have been developed to improve the specificity of Cas nucleases[2]. Many approaches for genome-wide identification of potential Cas-nuclease-induced double-strand breaks (DSBs) have been developed[3–18]. They can be generally divided into two categories: cell-free and cell-based methods (Supplementary Data 1). Particularly, the cell-free-based method identified off-target sites in vitro, such as Digenome-seq[3], SITE-Seq[4], and CIRCLE-seq[5]/CHANGE-seq[6], etc., they use a purified genome DNA as the targeted reference, which bypass efficient cellular transduction or transfection and avoid cell fitness effects. However, it is prone to obtaining false-positive results because the cellular properties, such as the chromatin and nuclear architecture, are not considerate.

In contrast, the cell-based strategies examine nuclease activity in vivo resulting in the detection of *bona fide* off-target sites, which improves the specificity of off-target events detection. The cell-based methods also can be classified into two groups: 1) direct method. It can directly reflect the DSBs at the moment of cell collection, such as BLESS[8]/BLISS[9], DSBCapture[10], and END-seq[11]. 2) indirect method. It identifies DSBs through obtaining DNA fragments bound by key factors in DNA repair processes by chromatin immunoprecipitation (ChIP) technologies, such as ChIP-seq[12] and DISCOVER-Seq[13]. On the other hand, IDLV capture[14], GUIDE-seq[15], and other GUIDE-seq-based methods[16,17] identify DSBs through recombination with an exogenous marker DNA. The direct methods always require multiple processing for library's construction, making it labor-intensive and impractical for analyzing large numbers of targets in parallel[8–11]. The indirect ChIP-based techniques require a high specificity of antibodies and display low-sensitivity[12,13]. Among the cell-based techniques, GUIDE-seq is the most widely used methods for identification of CRISPR-Cas nuclease induced off-target sites, it achieves the high sensitivity through detecting the accumulation of integrated double-stranded oligodeoxynucleotide (dsODN) at break sites in living cells over time[15]. However, in the original GUIDE-seq method its presented workflow is relative high cost and time-consuming (Supplementary Table 1). Besides, GUIDE-seq data analysis requires four essential files, Read1/Read2 and Index1/Index2 (contained the unique molecular index (UMI) and the sample barcode) for the DSB sites identification[15,19], making it get more complex. This greatly limits its broad applications. Therefore, a more simple and universal method for genome-wide specificity assessment of CRISPR/Cas nucleases is essential for the genome-editing research.

Here, we present Tag-seq, an improved method based on GUIDE-seq, by optimizing the donor DNA, adapters, PCR primers, and the library-preparation procedures to generate a simple and convenient platform. It can be broadly adopted for genome-wide DSBs detection with only laboratory routine reagents and commercially available high-throughput sequencing platforms. We also provide a comprehensive pipeline for implementing the analysis, which is available online at https://github.com/zhoujj2013/Tag-seq and https://doi.org/10.5281/zenodo.4679460.

## Results and discussion

**Tag-seq: a simple method for genome-wide assessment of DSBs**. Tag-seq aims to simplify the experimental pipeline and provide an easy alternative for genome-wide assessment of DSBs. Thus, similar to GUIDE-seq, Tag-seq was designed to profile genome-wide DSBs with an optimized double-strand DNA (termed Tag). It involves five steps: cell transfection, genomic DNA extraction, a single-tube reaction (including fragmentation, end repair, dA-tailing, and adapter ligation), PCR amplification, and sequencing (Fig. 1a). Then, we developed a bioinformatic analysis for DSBs sites detection (Fig. 1b, c). The substantial improvements of the Tag-seq workflow are listed as following: 1) a single-tube reaction method is applied in molecular manipulation process, including DNA fragmentation, end repair, dA-tailing and adapter ligation, which greatly shorten the time for libraries preparation (Fig. 1a); 2) the sample barcodes and UMI used in Tag-seq are compatible with commercial sequencing devices (Supplementary Fig. 1a); 3) a GC content balanced (45.7%) oligonucleotide Tag and a polyetherimide (PEI)-based transfection method enable high integration of Tag in DSB sites with cost-efficiency (Supplementary Fig. 1b–e and Supplementary Fig. 2); 4) Tag-specific primers with the "GAT" and "CA" nucleotide motifs ensure a high fidelity of PCR amplification (Supplementary Fig. 1a)[16]; 5) the all-in-one PCR strategy skipping the cleanup step[15] enables time-saving during libraries preparation (Supplementary Fig. f–h); 6) A state of art bioinformatic analysis pipeline is developed for Tag-seq analysis, it provides 3 visualization modules for inspecting the distribution of DSBs in a genome-wide level, the editing frequencies of a specific DSB and the comparison of a specific DSB among Tag-seq experiments (Fig. 1b, c). In summary, Tag-seq provides an easy alternative way for DSBs identification through introducing less costs, less time- and effort-consuming procedures and a comprehensive bioinformatic analysis workflow (Fig. 1 and Supplementary Table 1).

**Tag-seq accurately identifies and characterizes Cas-induced DSBs in a genome-wide level**. After we established Tag-seq protocol, we intended to test the performance of Tag-seq in identifying the Cas-induced DSBs. As a result, we found that Tag-seq with transfection using Lonza electroporation method obtained more reads at *EMX1*, *PD1,* and *CTLA4* loci, especially the on-target sites (Fig. 2a–c), indicating a higher efficiency compared to PEI-based approach. And we also noted that the Lonza-based method displayed less off-targets in all 3 tested sgRNAs. We speculated the possible reasons were that Lonza method was a nucleofector electroporation, which can fast mediate plasmids into the nuclei and lead to efficient expression, while the PEI method was a common transfection required a long time of process including cellular uptake, nuclear trafficking, and subcellular retention[20]. However, PEI is a routine reagent in molecular laboratory and it is quite cheap. Therefore, PEI-base method provides an easy and cost-efficient alternative for Lonza method. Next, we tested whether Tag-seq can identify off-target sites in different cell types. Expectedly, *As*Cpf1 targeting Site 6 and *Sp*Cas9 targeting *EMX1*, HEK293-Site1 and HEK293-site3 loci (four well-tested sites) showed that the cleavage events can be efficiently detected by Tag-seq whatever in HEK293T or MCF7 cell line (Fig. 2d, e; Supplementary Figs. 3 and 4a, b), and most of the detection sites were consistent with the previous reports[15,21] (the indels were verified by deep-seq at some sites, Supplementary Fig. 4a, b). Our analyses exhibited that Tag-seq recovered most of off-target sites in the tested sgRNAs. However, its ability in identifying the molecular features of DSBs remained unknown. We then sought to examine multiple target sites induced by *Sp*Cas9 and Cpf1 using Tag-seq. As a result, we observed that Tag-seq signal well displayed the cutting features of the Cas nucleases. In Tag-seq experiments on *Sp*Cas9, the signal in protospacer flanking regions showed a precise blunt cleavage at 3–4 bp upstream the NGG protospacer adjacent motif (PAM) (Supplementary Fig. 4c–e) and 1-base pair overhangs was found at *EMX1* protospacer because of the overlapping reads at this cutting site (including the on-target and off-target cleavages, Fig. 2f,

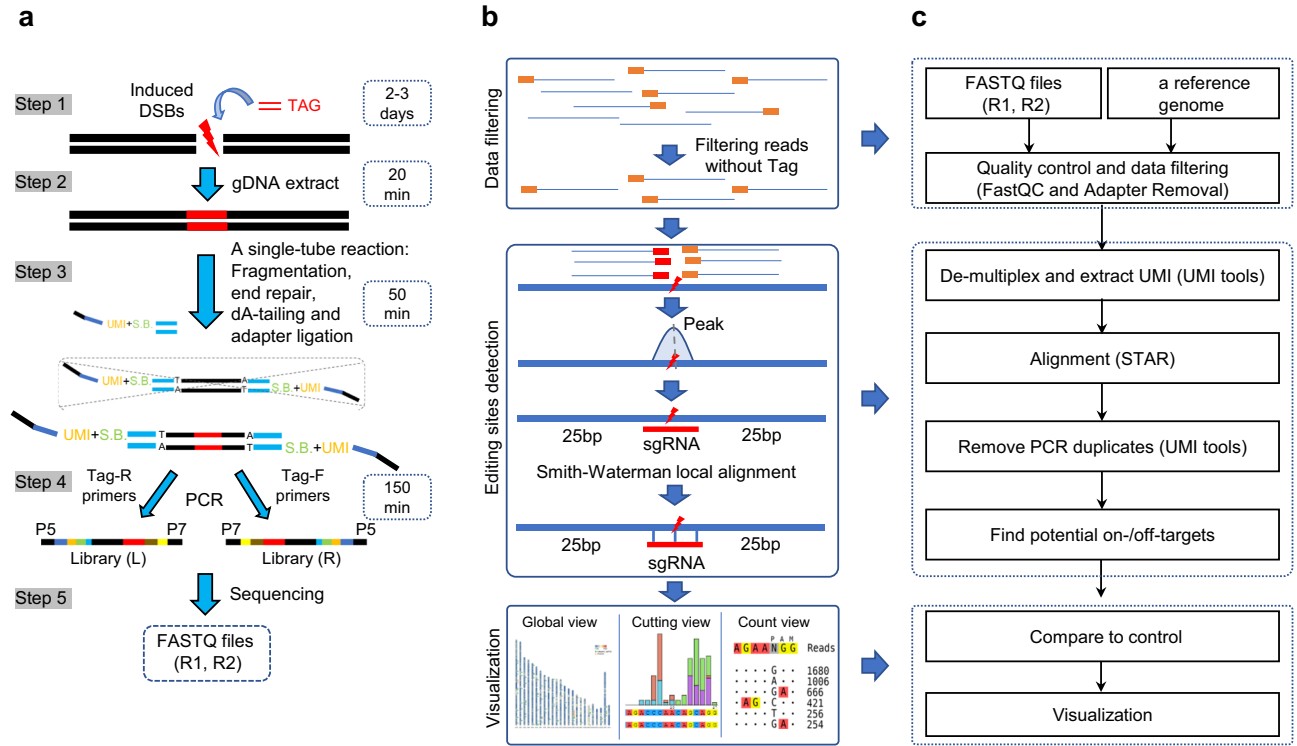

**Fig. 1 Overview of Tag-seq and its bioinformatics analysis workflow. a** Schematic of Tag-seq. It involves five steps: cell transfection, genomic DNA extraction, a single-tube reaction (including fragmentation, end repair, dA-tailing and adapter ligation), PCR amplification, and next-generation sequencing. It spends ~2–3 day + 220 min for libraries preparation. **b** The Tag-seq data analysis workflow contains three parts: data filtering, editing sites detection, and visualization. **c** Tag-seq data analysis scheme. It starts with only FASTQ files (R1, R2) and a reference genome.

g). The observations were consistent with previous in vitro[22] and in vivo studies[13]. Meanwhile, the signal of Cpf1 displayed multiple overhangs in *As*Cpf1 and *Lb*Cpf1 in both HEK293T and MCF7 cells (Fig. 2h, I and Supplementary Fig. 4f), confirming that Cpf1 tends to generate staggered double strand breaks when it cuts the target DNA[23]. These results demonstrated that Tag-seq accurately identifies Cas-induced DSBs in a genome-wide level and inspects the cellular activity of Cas nuclease-induced cleavages at a molecular level (Fig. 2).

**Tag-seq identifies off-target cleavages induced by Cas nuclease in a large-scale manner.** High-throughput measuring the off-target effects at diverse sites in a single transfection is efficient for evaluating the specificity of a nuclease on multiple sites[17]. To test this property of Tag-seq, we respectively performed 31 sgRNA targeted to 25 genes using *Sp*Cas9, and performed 23 sgRNA targeted to 12 genes using *As*Cpf1 in HEK293T cells. As a result, Tag-seq showed that all the potential off-target sites of the tested sgRNAs can be parallelly profiled by a single transfection (Figs. 3 and 4, and Supplementary Figs. 5 and 6), indicating that Tag-seq is a convenient platform for assessment the specificity of CRISPR/Cas nucleases in a scalable manner.

**Tag-seq successfully assess the specificity of CRISPR-Cas nuclease.** An aim of developing Tag-seq was to profile the specificity of the CRISPR/Cas systems. Therefore, the specificity of wild-type (WT) *Sp*Cas9 and two high-fidelity *Sp*Cas9 mutants, e*Sp*Cas9[24] and *Sp*Cas9-HF[25] targeted sites *EMX1*, *AAVS1*, and *CTLA4* were assessed. Consistent with previous reports, Tag-seq confirmed that e*Sp*Cas9 and *Sp*Cas9-HF exhibited a comparable activity and higher specificity than WT *Sp*Cas9, with detection of fewer or even no off-targets at these three tested sites (Fig. 5a–c).

Furthermore, Tag-seq also demonstrated that *As*Cpf1 was of higher specificity, while with lower activity than *Lb*Cpf1 at *CCR5* site (Fig. 5d), which was consistent with previous study at other sites[21]. Together, these results demonstrated that Tag-seq is an efficient method for assessing the specificity of Cas nucleases.

**Tag-seq discover integration sites of exogenous genes induced by transposons.** Genome-wide profiling the integrated sites in chromosomes is very important for safety assessment of the gene therapy approaches, such as viruses-based techniques or transposase-mediated tools[26]. The Sleeping Beauty (SB) transposase system is an efficient non-viral gene transfer tool, which can efficiently induce specific sequences of DNA inserting into genomes[27]. To extend the application of Tag-seq for mapping the insertion loci of the exogenous DNA induced by transposase, we constructed the WT *Sp*Cas9 and the nuclease-dead *Sp*Cas9 mutant fusing with SB transposase (termed Cas9-SB and dCas9-SB, respectively). Because we found that they can mediate efficient insertion and expression of enhanced green fluorescent protein (EGFP) in HEK293T cells (Fig. 6a and Supplementary Fig. 7a, b). Then, we applied Tag-seq to profile the Cas9-SB- and dCas9-SB-meditated EGFP insertion sites using the SB-transposon-arm-specific primer instead of the Tag sequence primers (Fig. 6b and Supplementary Fig. 7c, d). As a result, the EGFP fragments were randomly inserted into intergenic and intronic regions, and the flanking sequences of the insertion sites were enriched with TA dinucleotide motif (Fig. 6c–e), which was consistent with the strict TA-preference of SB-mediated integrations[28,29]. Thus, Tag-seq not only can assess the specificity of CRISPR-Cas nuclease but also discover the locations of exogenous gene integration in genome induced by transposase in an unbiased manner.

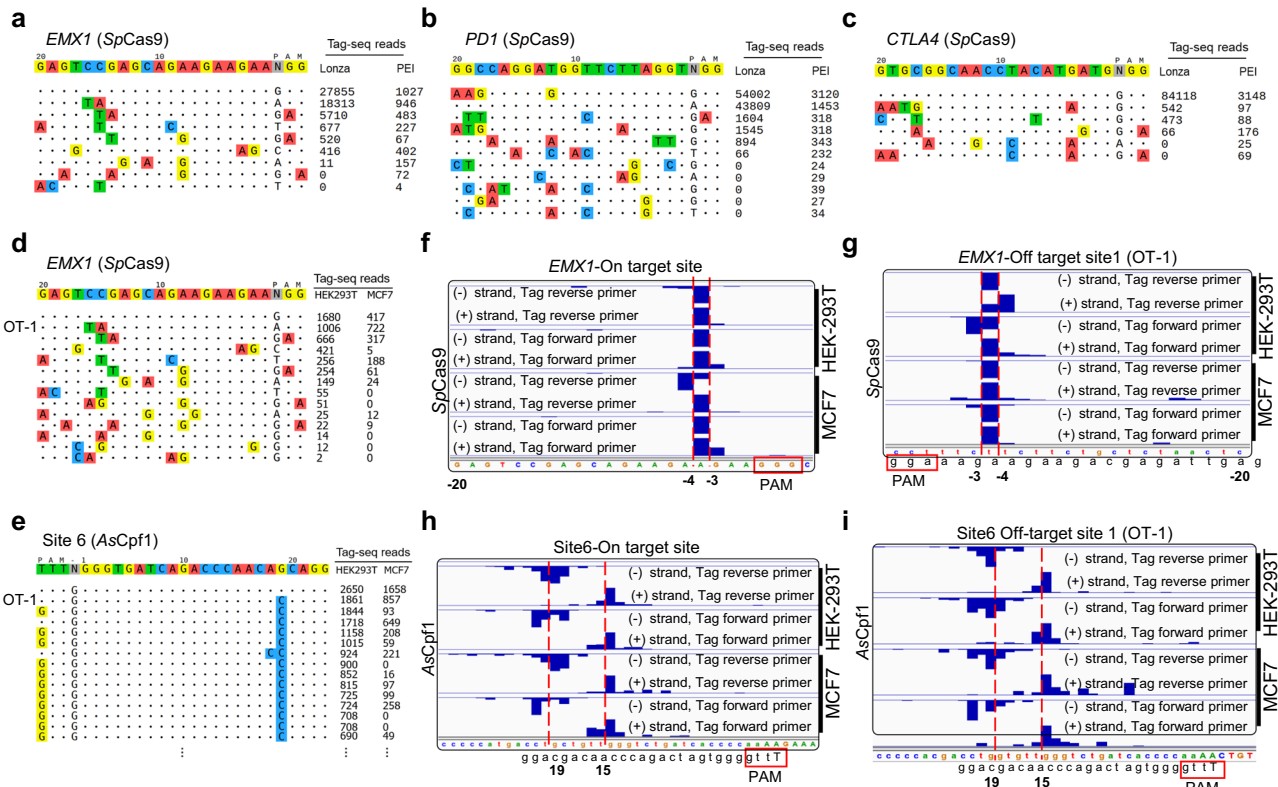

**Fig. 2 Tag-seq global profiles and molecular characterizes off-target cleavages induced by Cas-nuclease. a–c** PEI-based transfection and electroporation (Lonza Nucleofection)-based methods displayed comparable results in detection of off-target sites at *EMX1*, *PD1*, and *CLTA4* loci. **d**, **e** Off-target sites detection with Tag-seq. *Sp*Cas9 and *As*Cpf1 targeting *EMX1* and Site 6, respectively, induced off-target cleavages in HEK293T and MCF7 cells. For visualization, the targeted sgRNA was shown in the first line and the on-target and the off-target sites were shown without or with mismatches to the sgRNA sequence by color highlighting. Sequencing read counts were shown to the right of each site. Due to the limited space, the off-target sites for Site 6 were not completed displayed, and a full list was displayed in Supplementary Fig. 3. **f–i** Tag-seq displayed the characteristics of the *Sp*Cas9/Cpf1-induced-DSBs. Cutting at 3-4 bp upstream from the NGG PAM site and resulting in 1-bp overhangs at *EMX1* on-target (**f**) and off-target site 1 (OT-1) (**g**) sites for *Sp*Cas9 in both HEK293T and MCF7 cells. Resulting multiple overhangs at Site 6 on-target (**h**) and off-target site 1 (OT-1) (**i**) sites for *As*Cpf1 in both HEK293T and MCF7 cells. Red dotted lines indicated the cutting sites.

Compared with other GUIDE-seq based methods, Tag-seq performs with distinguished properties: 1) Tag-seq is an easy alternative for DSB identification, Tag-seq uses the classical nucleases for library construction in one step, which is rapid, easily available and efficient. Meanwhile, the preparation of libraries in tagmentation-based tag integration site sequencing (TTISS) is simplified by using the Tn5[17,30]. However, commercial Tn5 is relatively expensive, and, in our experience, purifying high-quality Tn5 in the lab, especially by the small groups, may present a challenge. 2) Tag-seq is more convenient. In Tag-seq, the UMI, sample barcode, adapter-genome ligation site, and Tag-genome integration site, which are all required to identify DSB sites, can be retrieved directly in the paired R1/R2 reads (Supplementary Fig. 1a) without the need of the additional index files. Therefore, Tag-seq is more suitable for the sequencing market niche and any vendor can do Tag-seq libraries sequencing, which is notably broad the potential user base for the Tag-seq technique, especially in terms of small labs and self-employed groups. (For more comparisons of the GUIDE-seq and GUIDE-seq-based methods, see Supplementary Table 2.)

## Conclusions

In summary, Tag-seq provides an alternative, complementary platform to conveniently and efficiently assess the specificity of CRISPR/Cas systems, the location of exogenous gene integration, or even to detect other DSBs induced by endogenous and exogenous factors.

## Methods

**Cell culture**. HEK293T and MCF7 cells were maintained in Dulbecco's Modified Eagle's Medium (DMEM, Life Technologies) and RPMI 1640 medium (Life Technologies) at 37 °C in a 5% CO$_2$ humidified incubator. All growth media were supplemented with 2 mM L-glutamine (Life Technologies), 100 U/mL penicillin, 100 μg/mL streptomycin (Life Technologies), and 10% fetal bovine serum. All the cell lines in this study were cultured no more than 10 passages.

**Cell transfection**. For detection of CRISPR off-target effects, HEK293T and MCF7 cells were transfected with PEI reagent (Polysciences, Inc., PA, USA) or Amaxa Cell Line Nucleofector Kit V (VCA-1003, Lonza, Switzerland) according to the manufacturer's instructions. Briefly, in the PEI transfection method, 250 ng of pCAG-Cas9-mcherry, 250 ng of FE-sgRNA-encoding plasmids[31], and 10 nM Tag/Oligo-1/Oligo-2 were transfected per well in a 24-well plate. In the Nucleofector Kit method, 1000 ng of pCAG-Cas9-mcherry, 1000 ng of FE-sgRNA-encoding plasmids, and 20 nM Tag were transfected per test. For large-scale off-target cleavages detection, HEK293T cells were transfected by PEI with 10 nM Tag, 1000 ng of *Sp*Cas9 or *As*Cpf1, and 1200 ng/1000 ng of total FE-sgRNA (31 for *Sp*Cas9 and 23 for *As*Cpf1) per well in a six-well plate. Cells were harvested 3 days after transfection and genomic DNA were extracted for library construction, sequencing, and performing bioinformatics analyses. For identification of insertion sites introduced by transposons, HEK293T cells were transfected by PEI with 250 ng of pCAG-dCas9/Cas-SB and 250 ng of donor EGFP-encoding plasmids per well in a 24-well plate. Cells were harvested 21 days for complete degradation of EGFP plasmid after a three-day puromycin selection starting the next day post-transfection. Then, genomic DNA were extracted for library construction, sequencing, and performing

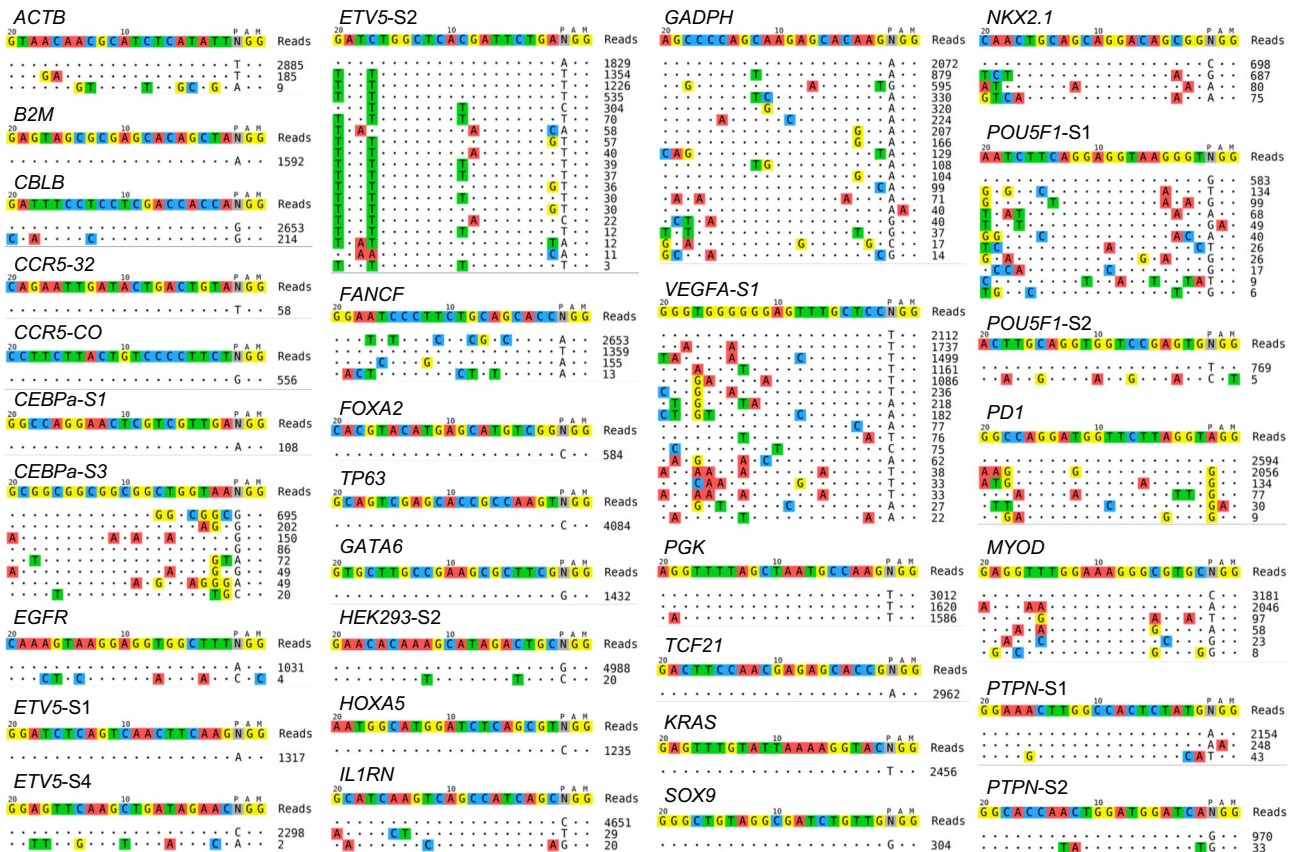

**Fig. 3 Tag-seq identifies off-target cleavages induced by Cas9-nuclease in a large-scale manner.** Tag-seq profiling the off-target sites induced by *Sp*Cas9 with 31 sgRNA in a single transfection. HEK293T cells were transfected with the donor Tag-oligo, *Sp*Cas9 and a pool mix of 31 sgRNA. Cells were harvested three days after transfection, and genomic DNA was extracted for library construction and then sequenced. All the off-target cleavages of the 31 sites can be parallelly profiled by Tag-seq.

bioinformatics analysis. The donor DNA sequences and the sgRNAs used in this study are shown in Supplementary Table 3 and Supplementary Data 2.

### Deep-seq library construction and data analysis

*Library construction*. The primers with forward and reverse indexes were used to amplify the genomic regions in the first-round PCR. Then, equal amounts of the first PCR products were mixed and subjected to a second round of PCR with the P5- and P7-containing primers to generate the sequencing libraries. Paired-end sequencing was performed on the Hiseq/NovaSeq devices (Novogene, Beijing, China). Indel frequency were calculated as the ratio of (read counts with indel sequence)/(total sequencing read counts). And the integration rate was calculated as the ratio of (read counts with donor DNA sequence)/(total sequencing read counts). Deep-seq primers are listed in Supplementary Table 4.

*Data analysis*. Demultiplexing: The paired-end reads were separated into files according to the barcode sequences of different samples.

Identification of potential inserted sequence: The demultiplexed reads were aligned to the genomic sequence of the targeted gene using BWA[32] (version 0.7.17) with default parameters. Then, the mapping boundary of each paired-end read were identified as a potential insertion.

Discrimination of Tag sequence: The Tag sequence was aligned with the region of insertions usingblast-short[33] (version 2.6.0) with adjusted parameters (-perc_identity 50 -evalue 0.01). Then, a Tag insertion was determined if a region of insertion with at least 16 bp matched a Tag sequence. Finally, the manner of insertion for different Tag sequences was determined according to the alignment result.

### Tag-seq library preparation and data analysis

Library construction Genomic DNA (gDNA) was purified using the TIANamp Genomic DNA Kit (TIANGEN Biotech Co., Ltd., Beijing, China), and then was fragmented, end repaired, dA-tailed, and ligated to adapters in a single tube with a Fragmentation, End Preparation, and dA-Tailing Module and Adapter Ligation Module kit (Vazyme Biotech Co., Ltd., Nanjing, China). To detect Cas-induced off-target events, a plus library (Tag forward primer, library R) and minus library (Tag reverse primer, library L) were generated by nested PCR with primers complementary to the Tag

sequence (two libraries per sample). To discover the insertion sites of EGFP mediated by transposons, libraries were generated by nested PCR with the transposon-arm-specific primer (one library per sample). Paired-end sequencing was commercially performed using HiSeq/NovaSeq (Novogene, Beijing, China). A detailed protocol is provided in the Supplementary Method, and all oligonucleotides and primers are listed in Supplementary Data 3.

*Data analysis*. Tag filtering: Demultiplexed raw reads were retained if they contain the Tag sequence at the beginning of the reverse read (second of pair, Read 2). The remaining paired-end reads were considered as derived from DSB sites induced by CRISPR/Cas9-derived RNA-guided nucleases (RGNs).

Quality control: The adapters and low-quality sequences were trimmed from the 3′ and 5′ ends. After trimming, reads shorter than 50 bp were discarded.

Read alignment: The remaining paired-end reads were aligned to the reference genome (hg19) using STAR 2.7.0c[34] with adjusted parameters (—alignIntronMax 50—outFilterScoreMinOverLread 0.5).

PCR duplicate consolidation: UMI-tools[35] (https://github.com/CGATOxford/UMI-tools) were applied to remove PCR duplicates. Briefly, reads that mapped to the same genomic position with the same UMI were considered to originate from the same pre-PCR molecule. Thus, those reads were consolidated into a single consensus read to improve the quantification of the Tag-seq signal. The consolidated alignment result was subjected to the identification of CRISPR RNA-guided nucleases (RGN)-mediated off-target cleavage sites or transposon-mediated integration sites.

### Identification of RGN-mediated off-target cleavage sites

The start mapping positions of reads amplified with the tag-specific primer (second of pair, Read 2) were converted to Browser Extensible Data (BED) format. The start mapping positions may vary because of random indel mutagenesis prior to non-homologous mediated end-joining, we then grouped the start mapping positions of reads into RGN-cutting hotspot regions if the distance among them was less than 10 bp. Furthermore, 10-bp sliding windows were created within the RGN-hotspot regions, and the read count of each sliding window in both the + and − strands of both the forward and reverse tag-specific libraries were calculated. Then, the signal.find_-peaks function from the SciPy ecosystem[36] (https://scipy.org/) was used to detect

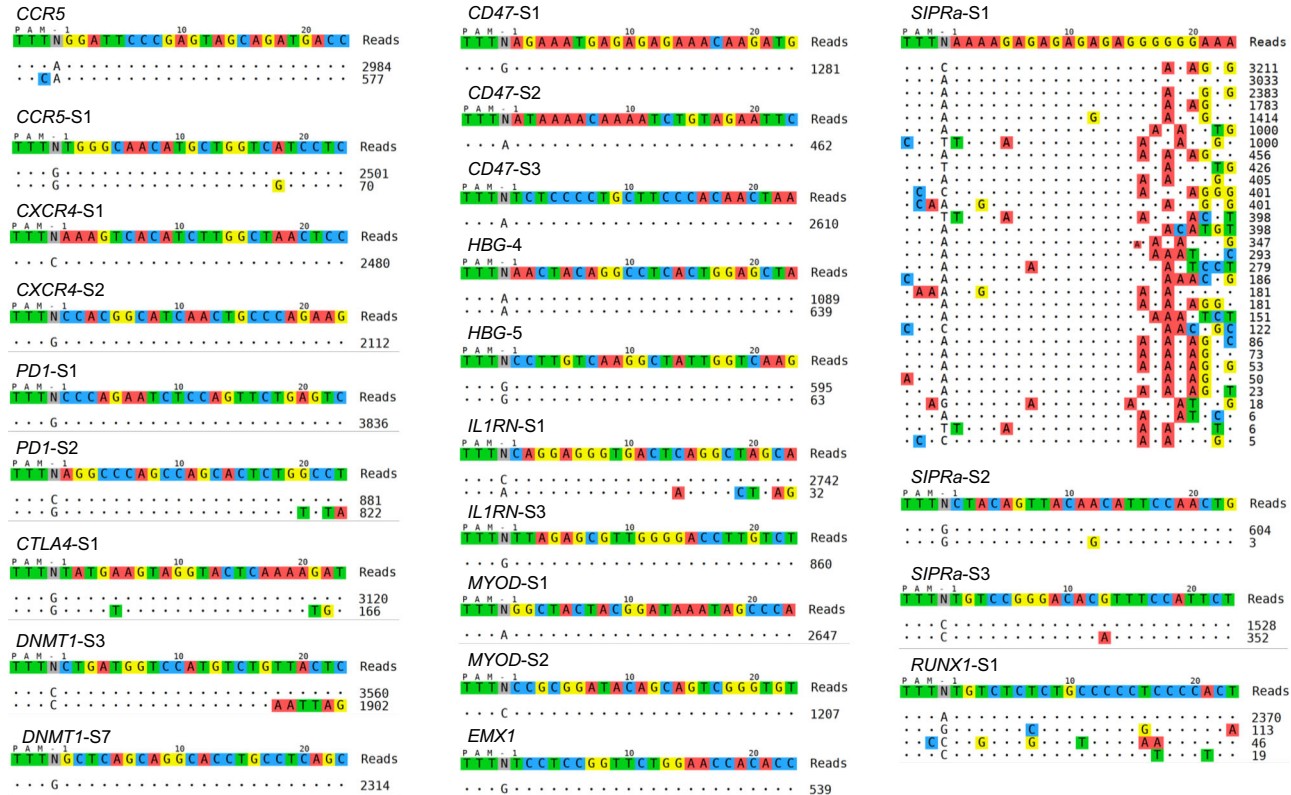

**Fig. 4 Tag-seq identifies off-target cleavages induced by Cpf1-nuclease in a large-scale manner.** Tag-seq profiling the off-target sites induced by *As*Cpf1 with 23 sgRNA in a single transfection. HEK293T cells were transfected with the donor Tag-oligo, *As*Cpf1 and a pool mix of 23 sgRNA. Cells were harvested three days after transfection, and genomic DNA was extracted for library construction and then sequenced. All the off-target cleavages of the 23 sites can be parallelly profiled by Tag-seq.

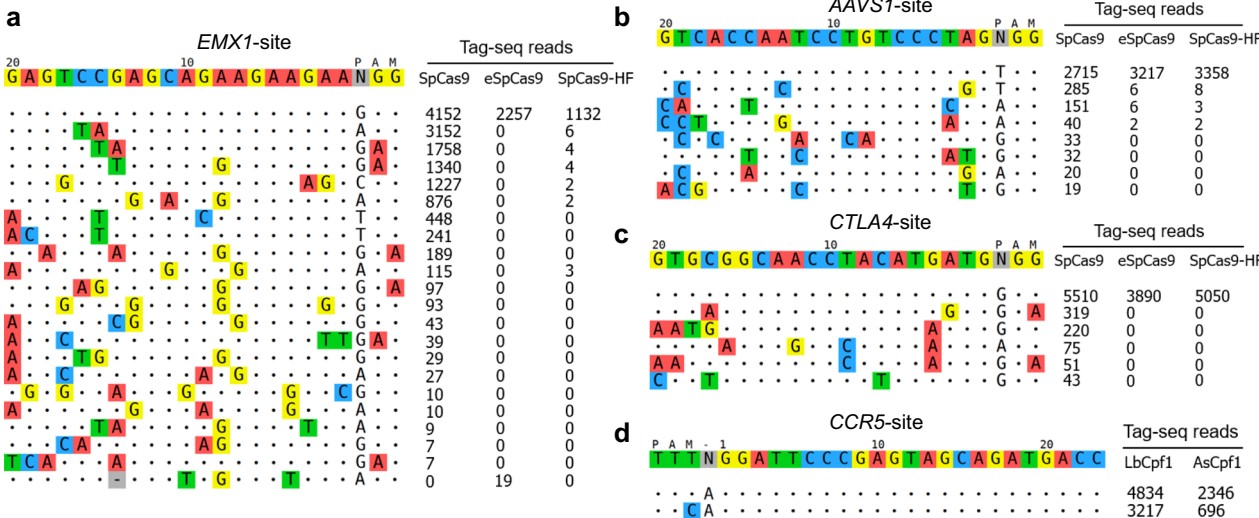

**Fig. 5 Tag-seq assesses the specificity of CRISPR. a–c** The specificity of wild-type *Sp*Cas9 and two high-fidelity variants (e*Sp*Cas9 and *Sp*Cas9-HF) were assessed at site *EMX1* (**a**), *AAVS1* (**b**), and *CTLA4* (**c**). **d** The specificity of the *As*Cpf1 and *Lb*Cfp1 nucleases were assessed at site *CCR5*.

potential DSB sites with sufficient supporting reads. The peaks with more than five reads mapping to both the + and − strand, or the same strand but amplified with both forward and reverse tag-specific primers, were flagged as sites of potential DSBs. The DSBs were identified as an on-target sites if the flanking regions (±25 bp) exactly matched the gRNA using a Smith–Waterman local-alignment algorithm, while they were identified as an off-target site if the flanking regions (±25 bp) matched gRNA with less than or equal than six mismatches. Finally, the identified off-targets, sorted by read count of Tag-seq, were annotated in a final output table and visualized as a PDF file.

**Identification of transposon-mediated integration sites**. A similar workflow described above was used to identify integration sites mediated by Cas9/dCas9-fused SB transposase. Briefly, we detected peaks of + and − strands in the library from a tag-specific primer with the Tag-seq analysis pipeline. Then, the peaks with at least five supporting reads were defined as potential DSBs, and 25 bp of the genomic sequence were retrieved on either side of these potential DSBs. We performed motif enrichment analysis on the retrieved sequences using the HOMER[37] package findGenomeMotif.pl and confirmed the TA motif in DSBs induced by transposons. The enriched motifs, sorted by *p* value, were annotated in a plain text file.

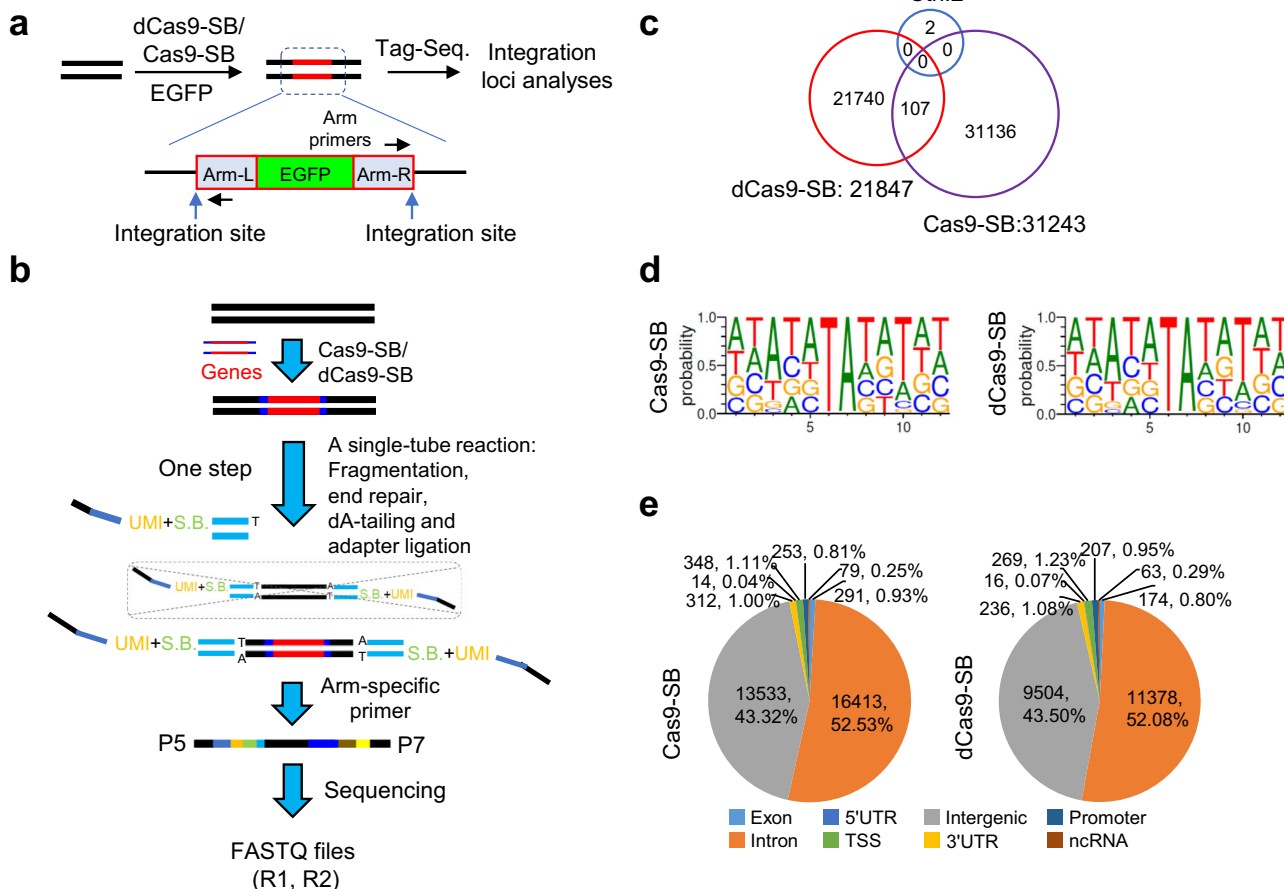

**Fig. 6 Tag-seq profiles the integration loci of exogenous genes induced by transposons. a** Diagram showing the use of Tag-seq to discover integration sites of EGFP induced by Cas9/dCas9 and Sleeping Beauty transposase fusion proteins. Arm-L/Arm-R, the left/right homology arm sequences. UMI, unique molecular index, S.B., sample barcode. **b** Workflow for construction of the library of Tag-seq for detection of the integration locations induced by Sleeping Beauty transposons. **c** Venn diagram showing the overlapping numbers of the insertion sites among the control, Cas9-SB, and dCas9-SB by Tag-seq. Ctrl, genomic DNA from blank-transfected cells was analyzed with Tag-seq; Cas9-SB/dCas9-SB, genomic DNA from cells transfected with the plasmids coding Cas9/dCas9-SB and the donor plasmid containing the SB-transposon-arm-flanked EGFP-expression cassette was analyzed with Tag-seq. **d** Sequence logos obtained via DNA sequences near the cleavage sites (±5 bp) identified by Tag-seq. **e** Distributions of the integrated locations in genome.

**Tag-seq data analysis pipeline**. A detailed description and the source code for Tag-seq data analysis pipeline are available at github (https://github.com/zhoujj2013/Tag-seq) and Zenodo (https://doi.org/10.5281/zenodo.4679460).

**FACS analysis**. FACS assays were performed for testing the EGFP insertion efficiency induced by transposase. HEK293T cells were harvested 21 days for complete degradation of EGFP plasmid after a three-day puromycin selection starting the next day post-transfection. EGFP expressing cells were gated by FITC channel and data were analyzed with the FlowJo software V7 (TreeStar, USA).

**Statistics analysis and reproducibility**. Student's t-test and One way ANOVA were used in this study for the statistical analysis. The reproducibility was showed by performing three independent biological replicate experiments.

**Reporting summary**. Further information on research design is available in the Nature Research Reporting Summary linked to this article.

## Data availability
All the sequencing data related to this study have been deposited in NCBI (Bioproject PRJNA678456). And other data that support the findings of this study are available from the corresponding author upon reasonable request. Besides, the source data for Supplementary Figs. 1d–e, and 5, 6 are shown in Supplementary Data 4.

## Code availability
All code for Tag-seq analysis pipeline in this study is available at https://github.com/zhoujj2013/Tag-seq and https://doi.org/10.5281/zenodo.4679460. Any updates will also be published on Zenodo and GitHub.

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

## Acknowledgements

We are grateful to all members of the Rong, Lin, and Zhou labs for helpful comments and discussions on the manuscript. This work was supported by the National Natural Science Foundation of China (82070002, 82072329, and 81872511), National Science and Technology Major Project (2018ZX10301101), Frontier Research Program of Bioland Laboratory (Guangzhou Regenerative Medicine and Health Guangdong Laboratory) (2018GZR110105005), the Natural Science Foundation of Guangdong Province (2018A030313455).

## Author contributions

Z.R., J.Z., Y.L., H.H. and Y.H. conceived the study, designed the experiments, analyzed the data, and wrote the manuscript. H.H. performed most experiments. J.Z. and Y.H designed the Tag-seq data analysis pipeline and analyzed all the sequencing data. G.H. performed deep-seq experiments. S.M. helped H.H. and Y.H. for some plasmids' construction. D.W. helped to the bioinformatics analysis. J.F. contributed to the editing of the manuscript.

## Competing interests

The authors declare no competing interests.
