## [Peer Review File · Communications Biology]

Reviewers' comments:

Reviewer #1 (Remarks to the Author):

Comments.

The authors developed the existing Guide-seq technology and presented the advanced Tag-seq method.

This methodology improves the efficiency of inserting dsODN tag into the DSB sites by optimizing the double stranded DNA tag. Tag-seq is generally improved in terms of cost, convenience, and accuracy than the existing Guide-seq technology through a simple sequencing method. In addition, it seems to be advantageous for multiplexing of various guide RNA induced off-target analysis or extending to other genome editing tools such as transposon conjugated CRISPR system. But, additional control experiments are required to show that the DSB capturing property is greatly improved compared to the existing Guide-seq methodology, otherwise, the Tag-seq's strengths will fade. And if my concerns are all addressed, I would recommend this paper for publication in Communications Biology.

Major concerns

- Optimization of Tag-seq methodology: A general experiment and explanation are needed to see if the amount of GC in the tag sequence affects the integration rate. It is necessary to explain how the improved tag works effectively during the process of insertion into the genome or the process of amplification.

- In Figure 2b, c: More gene targets are required to compare the integration rate(%) with Guide-seq method. Error bar indication and trial numbers (N=x) are also required.

- In Figure 3: Is there any other cell lines (e.g. MCF7) tested? For the reproducibility of the Tag-seq experiment, it is recommended to perform with more cell lines.

- In Figure 4, 5, 6: Guide-seq control is missing and the data is required to validate their claims. Multiplexing or other editing methods are also possible with Guide-seq, and direct comparison experiments with Guide-seq are required to show the superiority of Tag-seq.

Minor concerns

Detailed explanations are missing in the figure legend.

In Figures legend 1a: It needs a description of P5/P7.

In Figures legend 2a: Detailed information is required (e.g. asterisks=?)

In Figures legend 6a: It needs a description of Arm-L/R, UMI and S.B.

Reviewer #2 (Remarks to the Author):

"Tag-seq: a simple and flexible method for genome-wide specificity assessment of CRISPR/Cas nucleases" by Huang et al. is a manuscript that outlines a streamlined protocol for guided nuclease off-target detection using GUIDEseq-based methods, focusing on a different transfection technique. The authors perform a series of experiments to demonstrate the similarity between using the "Tag", rather than the dsODN developed for GUIDEseq, with their PEI-based transfection method.

It is claimed multiple times that the Tag-seq protocol is more cost effective, has reduced turnaround time, and offers the user more flexibility in comparison to GUIDEseq. The first of these claims mainly focuses on equipment differences for GUIDEseq-based assays (Lonza electroporator and Illumina sequencing platforms) but is subjective. The authors don't point out the differences in capital startup costs between the different Illumina instruments, or the cost per run. Rather their claim is based solely on the overall cost per base, which is dependent on the sample throughput of the user and the desired sequencing depth per sample. There is no mention of the sequencing depth or methods used to control sequencing depth effects from sample to sample even though the authors state that "higher sequencing depth results to higher sensitivity" with regards to

GUIDEseq-based assays. Additionally, any library with a P5/P7 Illumina adapter sequences can be sequenced on Illumina platforms (ie. MiSeq, HiSeq, NovaSeq), therefore the GUIDEseq protocol is not limited to the sequencing libraries on the MiSeq, but it is the laboratories choice to use the MiSeq rather than other Illumina platforms. The second claim, in regard to reduced turnaround time appears well warranted, yet time in their own protocols vary between 3 and 23 days between transfection and genomic DNA harvesting, indicating a large variability associated with the overall time to reach data. The third claim offering increased flexibility is not so clearly defined. For instance, if the flexibility suggests different transfection methods, the authors demonstrate the GUIDEseq can also be done with PEI-based transfection methods. Other authors evaluating improvements to GUIDEseq additionally indicate other areas in the protocol where GUIDEseq can be flexible (ie. post PCR1 cleanup seems optional). If the increased flexibility relates to use with the sleeping beauty transposon site detection, then this was not evaluated with GUIDEseq, only Tag-seq, so a comparison cannot be made.

The experimental results indicate similar performance between the GUIDEseq results and Tag-seq results, but do not investigate or speculate on the differences. For example, 14 additional sites were detected using Tag-seq versus GUIDEseq when assessing performance. This is claimed as an increase in sensitivity, but there is no validation that these new sites are actual off-targets or why GUIDEseq may have missed them. Rather, this could indicate a decrease in specificity or could simply be an imbalance between sequencing depth between the two data sets. Additionally, there is no assessment of significance presented. For example, is the sensitivity increase significantly greater than GUIDEseq at the same specificity? There are other examples similar to the above, i.e. specificity assessment. The Sleeping Beauty transposon integration site detection does not offer a quantitative comparison to previously identified site distributions, leaving open questions about accuracy.

Any source code used for processing and analysis should be archived on a platform that will support the code seemingly indefinitely. GitHub is a strong platform for development, but not for persisting code. Should the user account be removed, the code will also be deleted. Zenodo is a common location for archiving source code and could be used as an appropriate alternative.

With the major claims in the manuscript not being clearly demonstrated, it is difficult to recommend this manuscript for publication. The results would be of interest to others in the community, but likely different conclusions would be drawn from the results. A reasoning for PEI-based transfection compared to electroporation (such as with the Lonza device) could be supported with a direct comparison between the two, making an argument for using PEI-based transfection methods for similar performance at reduced cost. This comparison is absent from the analysis though, as GUIDEseq was seemingly evaluated with the PEI-based transfection method. Additional changes to the GUIDEseq-based protocol have been introduced by other publications, making the differences between Tag-seq and other GUIDEseq-based methods fewer than the difference between Tag-seq and GUIDEseq. Novel changes in the Tag-seq protocol include the transfection method and a streamlined one-step fragmentation/end-repair/dA-tailing/ligation. Other changes have been proposed in other manuscripts referenced by the authors.

Response to Reviewer #1:

The authors developed the existing Guide-seq technology and presented the advanced Tag-seq method. This methodology improves the efficiency of inserting dsODN tag into the DSB sites by optimizing the double stranded DNA tag. Tag-seq is generally improved in terms of cost, convenience, and accuracy than the existing Guide-seq technology through a simple sequencing method. In addition, it seems to be advantageous for multiplexing of various guide RNA induced off-target analysis or extending to other genome editing tools such as transposon conjugated CRISPR system. But, additional control experiments are required to show that the DSB capturing property is greatly improved compared to the existing Guide-seq methodology, otherwise, the Tag-seq's strengths will fade. And if my concerns are all addressed, I would recommend this paper for publication in Communications Biology.

Major concerns:

1. Optimization of Tag-seq methodology: A general experiment and explanation are needed to see if the amount of GC in the tag sequence affects the integration rate. It is necessary to explain how the improved tag works effectively during the process of insertion into the genome or the process of amplification.

Thanks for your suggestions. To test whether the amount of GC in the donor templates affects the integration rate, we have designed additional 2 extreme oligonucleotides with a low GC content of 11% and a high GC content of 74% and compared their efficiency in inserting into the genomic DNA. As shown in main text Page 5, line 112-114 and Supplementary Fig. 1b, e, the Tag with a balanced GC content (~46%) is with higher integration rate than other 2 extreme oligonucleotides in 5 tested sites. In addition, the improved Tag with a GC content of 40%-60% is an ideal template to design primers for PCR amplification[1]. However, we do not know how the Tag works effectively during the process of insertion into the genome, which requires further investigations.

2. In Figure 2b, c: More gene targets are required to compare the integration rate(%) with Guide-seq method. Error bar indication and trial numbers (N=x) are also required.

Thanks for your useful suggestions. Because there are some difficulties for us to do a side-by-side comparison between GUIDE-seq and Tag-seq and the reasons can be found in the following Point 4. Thus, we have to remove any statements comparing GUIDE-seq and Tag-seq performance, such as the comparison of integration rate for the dsODN and the Tag. However, we have performed additional assays to see how the GC content affects the integration rate by comparing with another

two oligo, oligo-1(GC=11%) and oligo-2 (GC=74%). As displayed in main text Page 5, line 112-114 and Supplementary Fig. 1b, e, three independent biological replicates results (mean values are presented with S.E.M, n=3) demonstrate that the Tag sequence can efficiently integrate into the DSB sites induced by CRISPR-Cas nucleases with variable level at different loci and show a higher integration rate than other 2 extreme oligonucleotides in 5 tested sites.

3. In Figure 3: Is there any other cell lines (e.g. MCF7) tested? For the reproducibility of the Tag-seq experiment, it is recommended to perform with more cell lines.

Thanks for helpful comments. Tag-seq accurately identifies and characterizes Cas-induced DSBs in both HEK293 and MCF7 cell lines using two common nucleases, SpCas9 and Cpf1, targeting to EMX1 and Site 6 loci, respectively. We have included those results in main text Page 6, line 139-143 and Fig. 2d, e; Supplementary Fig. 2.

4. In Figure 4, 5, 6: Guide-seq control is missing and the data is required to validate their claims. Multiplexing or other editing methods are also possible with Guide-seq, and direct comparison experiments with Guide-seq are required to show the superiority of Tag-seq.

We thank for your constructive and helpful comments. We agree that the GUIDE-seq controls are required in Fig. 4, 5, 6 and this will be a significant improvement for our work to prove the superiority of Tag-seq. Thus, we try calling all the available companies that provide sequencing service in China and finally find out that it is still impossible to do GUIDE-seq due to the lack of a commercial MiSeq sequencing provider.

*In theory, GUIDE-seq can be sequenced by MiSeq/NextSeq/HiSeq/NovaSeq devices. However, previous studies showed that GUIDE-seq libraries almost only be sequenced by MiSeq and NextSeq, which is a low-throughput and relative high cost sequencing platform in commercial (see following **Table R1** for the publications that employed GUIDE-seq method and their sequencing platforms). In our situation, we cannot find a vendor for GUIDE-seq library sequencing. Because the GUIDE-seq data analyses requires R1/R2 and Index1(I1)/Index2(I2) files[2, 3] (see following **Fig. R1** for more details for GUIDE-seq and Tag-seq methods). Additional procedures are required to generate I1/I2 files [2]. We had called all the available sequencing companies in China. Only one reliable company retain MiSeq platform, however, commercial companies usually provide service in a manner of stream-line with only sequencing R1 and R2 files without providing I1/I2 files, and they refused to change their fixed configuration for a specific customer. The company staff also told us*

they are closing MiSeq service since more and more customers switch to HiSeq and NovaSeq platforms. MiSeq platform can't make money any more because they are low-throughput. Therefore, for the scientists who own sequencing machines by themselves, it is possible to do GUIDE-seq. However, for other scientists without their own machines (like us), it is impossible to persuade a vendor to do customized GUIDE-seq. And there are the reasons that we are unable to do side-by-side comparison between GUIDE-seq and Tag-seq.

Since it's difficult for us to do the GUIDE-seq, it is very regrettable that the comparison results are unable to be presented in the current manuscript. Thus, we have to revise the manuscript with removing the statements comparing the performance of GUIDE-seq and Tag-seq. Actually, the original intention of our work is not to compare Tag-seq with GUIDE-seq but provide an easy alternative for GUIDE-seq, which can perfectly fit the sequencing market niche and any vendor can do Tag-seq sequencing. This is actually a huge advantage of Tag-seq over GUIDE-seq.

Table R1. The sequencing platform used in the GUIDE-seq.

References	Year	Applications	Sequencing platform
[3]	2015	Original GUIDE-seq	Illunima (MiSeq)
[4]	2015	Specificity Analysis(SaCas9)	Illunima (MiSeq)
[5]	2015	Specificity Analysis(SpCas9)	Illunima (MiSeq)
[6]	2015	Specificity Analysis(SaCas9)	Illunima (MiSeq)
[7]	2015	Specificity Analysis(SpCas9)	Illunima (MiSeq)
[8]	2016	Specificity Analysis(SpCas9)	Illunima (MiSeq)
[9]	2016	Specificity Analysis(Cas12a)	Illunima (MiSeq)
[10]	2016	Specificity Analysis(SpCas9)	Illunima (MiSeq)
[11]	2017	Specificity Analysis(SpCas9)	Illunima (MiSeq)
[12]	2017	Specificity Analysis(SpCas9)	Illunima (MiSeq)
[13]	2017	Specificity Analysis(SpCas9)	Illunima (MiSeq)
[14]	2017	Specificity Analysis(SpCas9)	Illunima (MiSeq)
[15]	2018	Specificity Analysis(NmeCas9)	Illunima (MiSeq)
[16]	2018	Specificity Analysis(SpCas9)	Illunima (MiSeq)
[17]	2018	Specificity Analysis(SpCas9)	Illunima (NextSeq 500/550)
[18]	2018	Specificity Analysis(SpCas9)	Illunima (MiSeq)
[19]	2018	Specificity Analysis(CBE)	Illunima (MiSeq)
[20]	2018	Specificity Analysis(SpCas9)	Illunima (MiSeq)
[21]	2018	Specificity Analysis(SpCas9)	Illunima (MiSeq)
[22]	2018	Specificity Analysis(CBE)	Illunima (MiSeq)
[23]	2018	Specificity Analysis(SpCas9)	Illunima (MiSeq)
[24]	2018	Specificity Analysis(SpCas9)	Illunima (MiSeq)
[25]	2018	Specificity Analysis(SpCas9)	Illunima (MiSeq)
[26]	2018	Specificity Analysis(SpCas9)	Illunima (MiSeq)
[27]	2019	Specificity Analysis(CBE)	Illunima (MiSeq)

[28]	2019	Specificity Analysis(Nme2Cas9)	Illunima (MiSeq)
[29]	2019	Specificity Analysis(AsCas12a)	Illunima (MiSeq)
[30]	2019	Specificity Analysis(SpCas9)	Illunima (MiSeq)
[31]	2019	vs IGUIDE	Illunima (MiSeq)
[32]	2019	Specificity Analysis(SpCas9)	Illunima (MiSeq)
[33]	2019	Specificity Analysis(CBE)	Illunima (MiSeq)
[34]	2019	Specificity Analysis(Cas12b)	Illunima (NextSeq)
[35]	2019	Specificity Analysis(Cas12b)	Illunima (MiSeq)
[36]	2019	Specificity Analysis(SpCas9)	Illunima (MiSeq)
[37]	2019	Specificity Analysis(ZFNs)	Illunima (MiSeq)
[38]	2019	Specificity Analysis(SpCas9)	Illunima (MiSeq)
[39]	2019	Specificity Analysis(SpCas9)	Illunima (MiSeq)
[40]	2019	vs DISCOVER-seq	Illunima (MiSeq)
[41]	2020	Specificity Analysis(SpCas9)	Illunima (NextSeq 500)
[42]	2020	Specificity Analysis(BlatCas9)	Illunima (MiSeq)
[43]	2020	vs CHANGE-seq	Illunima (NextSeq 500)
[44]	2020	Specificity Analysis(SpCas9)	Illunima (MiSeq)
[45]	2020	Specificity Analysis(SpCas9)	Illunima (NextSeq 550)
[46]	2020	Specificity Analysis(SpCas9)	Illunima (MiSeq)
[47]	2020	Specificity Analysis(SpCas9)	Illunima (MiSeq)
[48]	2020	Specificity Analysis(SpCas9)	Illunima (MiSeq)
[49]	2020	Specificity Analysis(SpCas9)	Illunima (MiSeq)
[50]	2020	Specificity Analysis(SpCas9)	Illunima (MiSeq)
[51]	2020	Specificity Analysis(SpCas9)	Illunima (MiSeq)
[52]	2020	Specificity Analysis(SpCas9)	Illunima (MiSeq)
[53]	2020	Specificity Analysis (SauriCas9)	Illunima (MiSeq)
[54]	2020	Specificity Analysis(SpCas9)	Illunima (NextSeq 500)
[55]	2020	Specificity Analysis(CeCas12a)	Illunima (MiSeq)
[56]	2020	Specificity Analysis(SpCas9)	Illunima (MiSeq)
[57]	2020	Specificity Analysis(LbCas12a)	Illunima (MiSeq)
[58]	2020	Specificity Analysis(SpCas9)	Illunima (MiSeq)
[59]	2020	Specificity Analysis(ABE)	Illunima (MiSeq)
[60]	2020	Specificity Analysis(SpCas9)	Illunima (MiSeq)
[61]	2020	Specificity Analysis(SpCas9)	Illunima (MiSeq)
[62]	2021	Specificity Analysis(SpCas9)	Illunima (MiSeq)

Fig. R1 Comparison between GUIDE-Seq and Tag-seq.

- (a) Workflow of the GUIDE-seq. It contained 8 major steps and spent approximately 2-3 days+365 min.
- (b) Schematic of the GUIDE-seq library^[3]. The library is flanked by P5 and P7 adaptors. Read1 retrieves genomic DNA sequence (R1). Read2 retrieves dsODN+genomic sequences (R2). Index1 seq Primer retrieves library index (I1). Index2 seq Primer retrieves sample barcode (SB, for demultiplexing) and unique molecular index (UMI, for removing the PCR duplicates) (I2). The unique feature of GUIDE-seq is that the SB and UMI are designed before the Read 1 primer and thus could not be retrieved directly in the Read1 files. Therefore, in GUIDE-seq Miseq instrument needs to be configured and perform an extra sequencing using the Index2 seq Primer to retrieve SB and UMI information (I2) ^[2].
- (c) Workflow of GUIDE-seq analysis. R1, R2, I1, and I2 are all required to analyze GUIDE-seq data.
- (d) Workflow of the Tag-seq. It contained 5 major steps and spent approximately 2-3 days+220 min.
- (e) Schematic of the Tag-seq library. The library is flanked by P5 and P7 adaptors. Read1 retrieves UMI, SB, and genomic DNA sequences (R1). Read2 retrieves Tag+genomic sequences (R2). Index1 seq Primer retrieves library index (I1). Library index is used to isolate reads from other libraries sequenced in the same lane.
- (f) Workflow of Tag-seq analysis. The sequencing vendors isolate data for each library from the bulk data in the same sequencing lane. For each Tag-seq library, only R1 and R2 are required for analysis.

***NOTE:** Sequencing vendors routinely used fixed configuration to sequence R1, R2, and I1 with MiSeq, HiSeq, and NovaSeq. No vendor would like to change machine configurations for a specific customer. For the scientists who own sequencing machines by themselves, it is possible to do GUIDE-seq. However, for other scientists without their own machines (like us), it is impossible to persuade a vendor to do GUIDE-seq. Therefore, we design Tag-seq library to fit the sequencing market niche and any vendor can do Tag-seq sequencing. This is actually a huge advantage of Tag-seq over GUIDE-seq.*

Minor concerns:

Detailed explanations are missing in the figure legend.

In Figures legend 1a: It needs a description of P5/P7.

In Figures legend 2a: Detailed information is required (e.g. asterisks=?)

In Figures legend 6a: It needs a description of Arm-L/R, UMI and S.B.

We thank reviewers for the helpful suggestions. We have revised the legends and descriptions as highlighted with red in the figure legends in the revised manuscript.

Response to Reviewer #2:

“Tag-seq: a simple and flexible method for genome-wide specificity assessment of CRISPR/Cas nucleases” by Huang et al. is a manuscript that outlines a streamlined protocol for guided nuclease off-target detection using GUIDEseq-based methods, focusing on a different transfection technique. The authors perform a series of experiments to demonstrate the similarity between using the “Tag”, rather than the dsODN developed for GUIDEseq, with their PEI-based transfection method.

It is claimed multiple times that the Tag-seq protocol is more cost effective, has reduced turnaround time, and offers the user more flexibility in comparison to GUIDEseq. The first of these claims mainly focuses on equipment differences for GUIDEseq-based assays (Lonza electroporator and Illumina sequencing platforms) but is subjective. The authors don't point out the differences in capital startup costs between the different Illumina instruments, or the cost per run. Rather their claim is based solely on the overall cost per base, which is dependent on the sample throughput of the user and the desired sequencing depth per sample. There is no mention of the sequencing depth or methods used to control sequencing depth effects from sample to sample even though the authors state that “higher sequencing depth results to higher sensitivity” with regards to GUIDEseq-based assays.

We thank the reviewer for advising the comparisons in capital startup costs between the different Illumina instruments, or the cost per run. We now have made a side-by-side comparison among the

procedures, times and costs between GUIDE-seq and Tag-seq method as displayed in main text Supplementary Table 2. As sequencing with the same depth (1 G raw data), the cost of the Tag-seq is much cheaper with approximately 33.5 USD/run, while the GUIDE-seq method is with greater than 150.5 USD/run.

Additionally, any library with a P5/P7 Illumina adapter sequences can be sequenced on Illumina platforms (ie. MiSeq, HiSeq, NovaSeq), therefore the GUIDEseq protocol is not limited to the sequencing libraries on the MiSeq, but it is the laboratories choice to use the MiSeq rather than other Illumina platforms.

*Thanks for your suggestions. In theory, GUIDE-seq can be sequenced by MiSeq/NextSeq/HiSeq/ NovaSeq. However, previous studies showed that GUIDE-seq libraries almost only be sequenced by MiSeq and NextSeq, which is a low-throughput and relative high cost sequencing platform in commercial (see **Table R1** for the publications that employed GUIDE-seq method and their sequencing platforms). This is because the GUIDE-seq data analyses requires R1/R2 and Index1(I1)/Index2(I2) files (see **Fig. R1** for more details for GUIDE-seq and Tag-seq methods). I1/I2 files generally need to additional procedures for generation[2]. Commercial companies usually provide service in a manner of stream-line with only sequencing R1 and R2 files without providing I1/I2 Reads. No vendor would like to change machine configurations for a specific customer. Thus, for the scientists who own sequencing machines by themselves, it is possible to do GUIDE-seq. However, for other scientists without their own machines (like us), it is impossible to persuade a vendor to do GUIDE-seq. Therefore, we design Tag-seq library to fit the sequencing market niche and any vendor can do Tag-seq sequencing. This is actually a huge advantage of Tag-seq over GUIDE-seq.*

The second claim, in regard to reduced turnaround time appears well warranted, yet time in their own protocols vary between 3 and 23 days between transfection and genomic DNA harvesting, indicating a large variability associated with the overall time to reach data.

Thanks for your suggestions. Generally, the genomic DNA collection in Tag-seq is 2-3 days post-transfection. We apologize to make reviewer misunderstanding in genomic DNA harvesting with a long time of 23 days. The reason is that we don't make it clearly in the manuscript and we appreciated for the reviewer's useful comments. Now we have revised this information clearly in the revised manuscript as highlighted with red in Page 11, line 251-256. In the experiments of the identification of the integration loci of exogenous genes induced by transposons, we transfected the WT SpCas9 and the nuclease-dead SpCas9 mutant fusing with SB transposase (termed Cas9-SB

and dCas9-SB, respectively) and the donor template DNA (a EGFP-expression cassette) plasmids into the HEK293T cells. Following, to demonstrate that the Cas9-SB and dCas9-SB can mediate the EGFP-donor efficient insertion and expression in HEK293T cells, we cultured the transfected cells with a time of 23 days to complete degradation of EGFP-donor plasmid for the more precise examination of the integration rate by FACS, and then we harvested these cells for libraries construction and integration locations identification.

The third claim offering increased flexibility is not so clearly defined. For instance, if the flexibility suggests different transfection methods, the authors demonstrate the GUIDEseq can also be done with PEI-based transfection methods. Other authors evaluating improvements to GUIDEseq additionally indicate other areas in the protocol where GUIDEseq can be flexible (ie. post PCR1 cleanup seems optional). If the increased flexibility relates to use with the sleeping beauty transposon site detection, then this was not evaluated with GUIDEseq, only Tag-seq, so a comparison cannot be made.

We thank the reviewer for giving such a good advice to improve the readability of our manuscript. Actually, "Tag-seq is flexible" means that it can be easily adapted to Miseq, HiSeq and Novaseq devices and sequenced within a single lane along with lots of other customers' samples without any special device configuration, while the GUIDE-seq need to do so[2, 3]. The UMI, sample barcode, adaptor-genome ligation site and Tag-genome integration site can be retrieved directly in the paired R1/R2 reads. We also develop a bioinformatics tool to analyze Tag-seq data with only paired R1/R2 reads (GUIDE-seq needs extra Index1 and Index2[2, 3]), and it is open and free. This is extremely simple and convenient. And in the revised version, to avoiding getting confused for the readers, we have removed it and replaced into "simple or convenient" in the main text.

The experimental results indicate similar performance between the GUIDEseq results and Tag-seq results, but do not investigate or speculate on the differences. For example, 14 additional sites were detected using Tag-seq versus GUIDEseq when assessing performance. This is claimed as an increase in sensitivity, but there is no validation that these new sites are actual off-targets or why GUIDEseq may have missed them. Rather, this could indicate a decrease in specificity or could simply be an imbalance between sequencing depth between the two data sets.

Thank you for the critical comments. As it is difficult to find a vender for GUIDE-seq libraries sequencing, we are unable to achieve the direct side-by-side comparisons between GUIDE-seq and Tag-seq. However, we still try to address this issue by using the off-target read data from the original GUIDE-seq article[3] at EMX1, HEK293-site1 and HEK293-site3, 3 sites had been well-studied in

GUIDE-seq[3] (Fig. R2). As a result, we find that Tag-seq could identify nearly all the GUIDE-seq-discovered sites (30/32) in these three tested sgRNAs, aside from two sites that had few reads supported in GUIDE-seq (only 5 at EMX1 and 2 at HEK293-site3). Interestingly, Tag-seq identifies another 14 new off-target sites. Furthermore, we choose 22 sites (including 14 sites are both detected in two methods and 8 sites are only detected in Tag-seq) among these 3 sites for validation using Deep-seq. As shown in Fig. R2, Deep-seq data demonstrate that the 14 sites detected in both methods are observed indels with level from 31.84% to 0.423%, while the rest 8 sites that only detected in Tag-seq also could be examined cleavages expect for 1 site with no detection in HEK293-site 1, assuring the sensitivity and specificity of Tag-seq. However, since these comparisons are not side-by-side achieved in our laboratory, thus this conclusion may present compromised.

Fig. R2 Comparisons of Tag-seq with GUIDE-seq.

(a-c) Sensitivity comparison between Tag-seq and GUIDE-seq at EMX1(a), HEK293-site1(b) and HEK293-site3(c), and some sites were verified deep-seq.

(d) Venn diagrams showing the number of off-target sites identified by Tag-seq and GUIDE-seq.

Note: GUIDE-seq off-target read data is from the original GUIDE-seq article[3], and the highlighted in red are the sites only detected in Tag-seq.

Additionally, there is no assessment of significance presented. For example, is the sensitivity

increase significantly greater than GUIDEseq at the same specificity? There are other examples similar to the above, i.e. specificity assessment. The Sleeping Beauty transposon integration site detection does not offer a quantitative comparison to previously identified site distributions, leaving open questions about accuracy.

Thanks for your suggestions. As the description above, we are unable to achieve GUIDE-seq, however, we still try to compare these two methods by using the off-target read data from the original GUIDE-seq article, and the results display Tag-seq can detect more off-target cleavages. Further, deep-seq verifies that these sites are the actual editing cases (Fig. R2), confirming that Tag-seq may be higher sensitive and specific than GUIDE-seq. In addition, for the detection of the Sleeping Beauty transposon integration sites, it have been reported that the insertion sites mediated by Sleeping Beauty (SB) system are strong with an AT motif preferences and the insertions are random on a genome-wide scale [63-66]. Although the inserted sites vary each time, our data demonstrated every insertion site induced by SB-transposon is accurately identified by Tag-seq with a strong AT motif (main text Fig. 5), which are consistent with the previous reports, confirming the accuracy of Tag-seq.

Any source code used for processing and analysis should be archived on a platform that will support the code seemingly indefinitely. GitHub is a strong platform for development, but not for persisting code. Should the user account be removed, the code will also be deleted. Zenodo is a common location for archiving source code and could be used as an appropriate alternative.

Thanks for your suggestion. We have presented the Tag-seq analyses pipeline not only in GitHub, but also in Zenodo, which is opened for free in the following addresses: <https://github.com/zhoujj2013/Tag-seq> and <https://doi.org/10.5281/zenodo.4679460>. We have added this content in main text Page 4, line 94-97; Page 13, line 314-317 and Page 15, line 349-351.

With the major claims in the manuscript not being clearly demonstrated, it is difficult to recommend this manuscript for publication. The results would be of interest to others in the community, but likely different conclusions would be drawn from the results. A reasoning for PEI-based transfection compared to electroporation (such as with the Lonza device) could be supported with a direct comparison between the two, making an argument for using PEI-based transfection methods for similar performance at reduced cost. This comparison is absent from the analysis though, as GUIDEseq was seemingly evaluated with the PEI-based transfection method.

Thanks for your useful suggestion. We have performed the comparison between PEI-based and

electroporation-base (Lonza kit) transfection in profiling the DSB sites detection by Tag-seq at EMX1, PD1 and CTLA4 sites. As shown in main text Fig. 2a-c, Lonza-base method obtains more reads at all tested 3 sites, especially at on-target sites, indicating a higher efficiency compared to PEI-based transfection. And we also note that the Lonza electroporation displays less off-targets in all 3 tested sgRNA. We speculate the possible reasons is that the Lonza electroporation is a nucleofector, which can mediate plasmids fast into the nuclei and efficient expression; While the PEI method is a common transfection required a process of cellular uptake, nuclear trafficking and subcellular retention[67]. However, PEI is a routine reagent in molecular laboratory and it is quite cheap (less 1 USD/test), while the Lonza method cost 25 times more than PEI. Therefore, PEI-base method provides an easy and cost-efficient alternative for Lonza electroporation. Besides, to do such electroporation transfection, it also needs a Lonza-special device, which is indeed not as convenient as the PEI-based approach, especially in terms of small labs and self-employed groups.

Additional changes to the GUIDEseq-based protocol have been introduced by other publications, making the differences between Tag-seq and other GUIDEseq-based methods fewer than the difference between Tag-seq and GUIDEseq. Novel changes in the Tag-seq protocol include the transfection method and a streamlined one-step fragmentation/end-repair/dA-tailing/ligation. Other changes have been proposed in other manuscripts referenced by the authors.

Thanks for your comments. Actually, the purpose of our work is not to compare Tag-seq with GUIDE-seq but provide an easy and cost-efficient alternative for GUIDE-seq, which can perfectly fit the sequencing market niche and any vendor can do Tag-seq sequencing. This is actually a huge advantage of Tag-seq over GUIDE-seq, especially in terms of small labs and self-employed groups (like us).

References

1. Dieffenbach CW, Lowe TM, Dveksler GS: **General concepts for PCR primer design.** *PCR Methods Appl* 1993, **3**(3):S30-37.
2. Tsai SQ, Topkar VV, Joung JK, Aryee MJ: **Open-source guideseq software for analysis of GUIDE-seq data.** *Nature biotechnology* 2016, **34**(5):483.
3. Tsai SQ, Zheng Z, Nguyen NT, Liebers M, Topkar VV, Thapar V, Wyvekens N, Khayter C, Iafrate AJ, Le LP *et al*: **GUIDE-seq enables genome-wide profiling of off-target cleavage by CRISPR-Cas nucleases.** *Nature biotechnology* 2015, **33**(2):187-197.
4. Friedland AE, Baral R, Singhal P, Loveluck K, Shen S, Sanchez M, Marco E, Gotta GM, Maeder ML,

- Kennedy EM *et al*: **Characterization of Staphylococcus aureus Cas9: a smaller Cas9 for all-in-one adeno-associated virus delivery and paired nickase applications.** *Genome biology* 2015, **16**:257.
5. Bolukbasi MF, Gupta A, Oikemus S, Derr AG, Garber M, Brodsky MH, Zhu LJ, Wolfe SA: **DNA-binding-domain fusions enhance the targeting range and precision of Cas9.** *Nature methods* 2015, **12**(12):1150-1156.
 6. Kleinstiver BP, Prew MS, Tsai SQ, Nguyen NT, Topkar VV, Zheng Z, Joung JK: **Broadening the targeting range of Staphylococcus aureus CRISPR-Cas9 by modifying PAM recognition.** *Nature biotechnology* 2015, **33**(12):1293-1298.
 7. Kleinstiver BP, Prew MS, Tsai SQ, Topkar VV, Nguyen NT, Zheng Z, Gonzales AP, Li Z, Peterson RT, Yeh JR *et al*: **Engineered CRISPR-Cas9 nucleases with altered PAM specificities.** *Nature* 2015, **523**(7561):481-485.
 8. Kleinstiver BP, Pattanayak V, Prew MS, Tsai SQ, Nguyen NT, Zheng Z, Joung JK: **High-fidelity CRISPR-Cas9 nucleases with no detectable genome-wide off-target effects.** *Nature* 2016, **529**(7587):490-495.
 9. Kleinstiver BP, Tsai SQ, Prew MS, Nguyen NT, Welch MM, Lopez JM, McCaw ZR, Aryee MJ, Joung JK: **Genome-wide specificities of CRISPR-Cas Cpf1 nucleases in human cells.** *Nature biotechnology* 2016, **34**(8):869-874.
 10. Yin H, Song CQ, Dorkin JR, Zhu LJ, Li Y, Wu Q, Park A, Yang J, Suresh S, Bizhanova A *et al*: **Therapeutic genome editing by combined viral and non-viral delivery of CRISPR system components in vivo.** *Nature biotechnology* 2016, **34**(3):328-333.
 11. Yin H, Song CQ, Suresh S, Wu Q, Walsh S, Rhym LH, Mintzer E, Bolukbasi MF, Zhu LJ, Kauffman K *et al*: **Structure-guided chemical modification of guide RNA enables potent non-viral in vivo genome editing.** *Nature biotechnology* 2017, **35**(12):1179-1187.
 12. Gomes-Silva D, Srinivasan M, Sharma S, Lee CM, Wagner DL, Davis TH, Rouce RH, Bao G, Brenner MK, Mamonkin M: **CD7-edited T cells expressing a CD7-specific CAR for the therapy of T-cell malignancies.** *Blood* 2017, **130**(3):285-296.
 13. Chen JS, Dagdas YS, Kleinstiver BP, Welch MM, Sousa AA, Harrington LB, Sternberg SH, Joung JK, Yildiz A, Doudna JA: **Enhanced proofreading governs CRISPR-Cas9 targeting accuracy.** *Nature* 2017, **550**(7676):407-410.
 14. Petris G, Casini A, Montagna C, Lorenzin F, Prandi D, Romanel A, Zasso J, Conti L, Demichelis F, Cereseto A: **Hit and go CAS9 delivered through a lentiviral based self-limiting circuit.** *Nat Commun* 2017, **8**:15334.
 15. Amrani N, Gao XD, Liu P, Edraki A, Mir A, Ibraheim R, Gupta A, Sasaki KE, Wu T, Donohoue PD *et al*:

- NmeCas9 is an intrinsically high-fidelity genome-editing platform.** *Genome biology* 2018, **19**(1):214.
16. Montagna C, Petris G, Casini A, Maule G, Franceschini GM, Zanella I, Conti L, Arnoldi F, Burrone OR, Zentilin L *et al*: **VSV-G-Enveloped Vesicles for Traceless Delivery of CRISPR-Cas9.** *Mol Ther Nucleic Acids* 2018, **12**:453-462.
 17. Nishimasu H, Shi X, Ishiguro S, Gao L, Hirano S, Okazaki S, Noda T, Abudayyeh OO, Gootenberg JS, Mori H *et al*: **Engineered CRISPR-Cas9 nuclease with expanded targeting space.** *Science* 2018, **361**(6408):1259-1262.
 18. Vakulskas CA, Dever DP, Rettig GR, Turk R, Jacobi AM, Collingwood MA, Bode NM, McNeill MS, Yan S, Camarena J *et al*: **A high-fidelity Cas9 mutant delivered as a ribonucleoprotein complex enables efficient gene editing in human hematopoietic stem and progenitor cells.** *Nature medicine* 2018, **24**(8):1216-1224.
 19. Gehrke JM, Cervantes O, Clement MK, Wu Y, Zeng J, Bauer DE, Pinello L, Joung JK: **An APOBEC3A-Cas9 base editor with minimized bystander and off-target activities.** *Nature biotechnology* 2018, **36**(10):977-982.
 20. Listgarten J, Weinstein M, Kleinstiver BP, Sousa AA, Joung JK, Crawford J, Gao K, Hoang L, Elibol M, Duench JG *et al*: **Prediction of off-target activities for the end-to-end design of CRISPR guide RNAs.** *Nat Biomed Eng* 2018, **2**(1):38-47.
 21. Wang L, Smith J, Breton C, Clark P, Zhang J, Ying L, Che Y, Lape J, Bell P, Calcedo R *et al*: **Meganuclease targeting of PCSK9 in macaque liver leads to stable reduction in serum cholesterol.** *Nature biotechnology* 2018, **36**(8):717-725.
 22. Yeh WH, Chiang H, Rees HA, Edge ASB, Liu DR: **In vivo base editing of post-mitotic sensory cells.** *Nat Commun* 2018, **9**(1):2184.
 23. Hu JH, Miller SM, Geurts MH, Tang W, Chen L, Sun N, Zeina CM, Gao X, Rees HA, Lin Z *et al*: **Evolved Cas9 variants with broad PAM compatibility and high DNA specificity.** *Nature* 2018, **556**(7699):57-63.
 24. Casini A, Olivieri M, Petris G, Montagna C, Reginato G, Maule G, Lorenzin F, Prandi D, Romanel A, Demichelis F *et al*: **A highly specific SpCas9 variant is identified by in vivo screening in yeast.** *Nature biotechnology* 2018, **36**(3):265-271.
 25. Yin H, Song CQ, Suresh S, Kwan SY, Wu Q, Walsh S, Ding J, Bogorad RL, Zhu LJ, Wolfe SA *et al*: **Partial DNA-guided Cas9 enables genome editing with reduced off-target activity.** *Nature chemical biology* 2018, **14**(3):311-316.
 26. Gao X, Tao Y, Lamas V, Huang M, Yeh WH, Pan B, Hu YJ, Hu JH, Thompson DB, Shu Y *et al*: **Treatment**

- of autosomal dominant hearing loss by in vivo delivery of genome editing agents.** *Nature* 2018, **553**(7687):217-221.
27. Webber BR, Lonetree CL, Kluesner MG, Johnson MJ, Pomeroy EJ, Diers MD, Lahr WS, Draper GM, Slipek NJ, Smeester BA *et al*: **Highly efficient multiplex human T cell engineering without double-strand breaks using Cas9 base editors.** *Nat Commun* 2019, **10**(1):5222.
28. Edraki A, Mir A, Ibraheim R, Gainetdinov I, Yoon Y, Song CQ, Cao Y, Gallant J, Xue W, Rivera-Perez JA *et al*: **A Compact, High-Accuracy Cas9 with a Dinucleotide PAM for In Vivo Genome Editing.** *Mol Cell* 2019, **73**(4):714-726 e714.
29. Maule G, Casini A, Montagna C, Ramalho AS, De Boeck K, Debyser Z, Carlon MS, Petris G, Cereseto A: **Allele specific repair of splicing mutations in cystic fibrosis through AsCas12a genome editing.** *Nat Commun* 2019, **10**(1):3556.
30. Gyorgy B, Nist-Lund C, Pan B, Asai Y, Karavitaki KD, Kleinstiver BP, Garcia SP, Zaborowski MP, Solanes P, Spataro S *et al*: **Allele-specific gene editing prevents deafness in a model of dominant progressive hearing loss.** *Nature medicine* 2019, **25**(7):1123-1130.
31. Nobles CL, Reddy S, Salas-McKee J, Liu X, June CH, Melenhorst JJ, Davis MM, Zhao Y, Bushman FD: **iGUIDE: an improved pipeline for analyzing CRISPR cleavage specificity.** *Genome biology* 2019, **20**(1):14.
32. Park SH, Lee CM, Dever DP, Davis TH, Camarena J, Srifa W, Zhang Y, Paikari A, Chang AK, Porteus MH *et al*: **Highly efficient editing of the beta-globin gene in patient-derived hematopoietic stem and progenitor cells to treat sickle cell disease.** *Nucleic acids research* 2019, **47**(15):7955-7972.
33. Carreras A, Pane LS, Nitsch R, Madeyski-Bengtson K, Porritt M, Akcakaya P, Taheri-Ghahfarokhi A, Ericson E, Bjursell M, Perez-Alcazar M *et al*: **In vivo genome and base editing of a human PCSK9 knock-in hypercholesterolemic mouse model.** *BMC Biol* 2019, **17**(1):4.
34. Strecker J, Jones S, Koopal B, Schmid-Burgk J, Zetsche B, Gao L, Makarova KS, Koonin EV, Zhang F: **Engineering of CRISPR-Cas12b for human genome editing.** *Nat Commun* 2019, **10**(1):212.
35. Kleinstiver BP, Sousa AA, Walton RT, Tak YE, Hsu JY, Clement K, Welch MM, Horng JE, Malagon-Lopez J, Scarfo I *et al*: **Engineered CRISPR-Cas12a variants with increased activities and improved targeting ranges for gene, epigenetic and base editing.** *Nature biotechnology* 2019, **37**(3):276-282.
36. Nadakuduti SS, Starker CG, Ko DK, Jayakody TB, Buell CR, Voytas DF, Douches DS: **Evaluation of Methods to Assess in vivo Activity of Engineered Genome-Editing Nucleases in Protoplasts.** *Front Plant Sci* 2019, **10**:110.
37. Paschon DE, Lussier S, Wangzor T, Xia DF, Li PW, Hinkley SJ, Scarlott NA, Lam SC, Waite AJ, Truong

- LN *et al*: **Diversifying the structure of zinc finger nucleases for high-precision genome editing.** *Nat Commun* 2019, **10**(1):1133.
38. Schirotti G, Conti A, Ferrari S, Della Volpe L, Jacob A, Albano L, Beretta S, Calabria A, Vavassori V, Gasparini P *et al*: **Precise Gene Editing Preserves Hematopoietic Stem Cell Function following Transient p53-Mediated DNA Damage Response.** *Cell stem cell* 2019, **24**(4):551-565 e558.
39. Pavel-Dinu M, Wiebking V, Dejene BT, Srifa W, Mantri S, Nicolas CE, Lee C, Bao G, Kildebeck EJ, Punjya N *et al*: **Gene correction for SCID-X1 in long-term hematopoietic stem cells.** *Nat Commun* 2019, **10**(1):1634.
40. Wienert B, Wyman SK, Richardson CD, Yeh CD, Akcakaya P, Porritt MJ, Morlock M, Vu JT, Kazane KR, Watry HL *et al*: **Unbiased detection of CRISPR off-targets in vivo using DISCOVER-Seq.** *Science* 2019, **364**(6437):286-289.
41. Chaudhari HG, Penterman J, Whitton HJ, Spencer SJ, Flanagan N, Lei Zhang MC, Huang E, Khedkar AS, Toomey JM, Shearer CA *et al*: **Evaluation of Homology-Independent CRISPR-Cas9 Off-Target Assessment Methods.** *CRISPR J* 2020, **3**(6):440-453.
42. Gao N, Zhang C, Hu Z, Li M, Wei J, Wang Y, Liu H: **Characterization of Brevibacillus laterosporus Cas9 (BlatCas9) for Mammalian Genome Editing.** *Front Cell Dev Biol* 2020, **8**:583164.
43. Lazzarotto CR, Malinin NL, Li Y, Zhang R, Yang Y, Lee G, Cowley E, He Y, Lan X, Jividen K *et al*: **CHANGE-seq reveals genetic and epigenetic effects on CRISPR-Cas9 genome-wide activity.** *Nature biotechnology* 2020.
44. Weber L, Frati G, Felix T, Hardouin G, Casini A, Wollenschlaeger C, Meneghini V, Masson C, De Cian A, Chalumeau A *et al*: **Editing a gamma-globin repressor binding site restores fetal hemoglobin synthesis and corrects the sickle cell disease phenotype.** *Sci Adv* 2020, **6**(7).
45. Chung CH, Allen AG, Atkins AJ, Sullivan NT, Homan G, Costello R, Madrid R, Nonnemacher MR, Dampier W, Wigdahl B: **Safe CRISPR-Cas9 Inhibition of HIV-1 with High Specificity and Broad-Spectrum Activity by Targeting LTR NF-kappaB Binding Sites.** *Mol Ther Nucleic Acids* 2020, **21**:965-982.
46. Rai R, Romito M, Rivers E, Turchiano G, Blattner G, Vetharoy W, Ladon D, Andrieux G, Zhang F, Zinicola M *et al*: **Targeted gene correction of human hematopoietic stem cells for the treatment of Wiskott - Aldrich Syndrome.** *Nat Commun* 2020, **11**(1):4034.
47. Basar R, Daher M, Uprety N, Gokdemir E, Alsuliman A, Ensley E, Ozcan G, Mendt M, Hernandez Sanabria M, Kerbauy LN *et al*: **Large-scale GMP-compliant CRISPR-Cas9-mediated deletion of the glucocorticoid receptor in multivirus-specific T cells.** *Blood Adv* 2020, **4**(14):3357-3367.
48. Lamsfus-Calle A, Daniel-Moreno A, Antony JS, Epting T, Heumos L, Baskaran P, Admard J, Casadei N,

- Latifi N, Siegmund DM *et al*: **Comparative targeting analysis of KLF1, BCL11A, and HBG1/2 in CD34(+) HSPCs by CRISPR/Cas9 for the induction of fetal hemoglobin**. *Scientific reports* 2020, **10**(1):10133.
49. Croci S, Carriero ML, Capitani K, Daga S, Donati F, Papa FT, Frullanti E, Lopergolo D, Lamacchia V, Tita R *et al*: **AAV-mediated FOXP1 gene editing in human Rett primary cells**. *Eur J Hum Genet* 2020, **28**(10):1446-1458.
50. Goodwin M, Lee E, Lakshmanan U, Shipp S, Froessl L, Barzaghi F, Passerini L, Narula M, Sheikali A, Lee CM *et al*: **CRISPR-based gene editing enables FOXP3 gene repair in IPEX patient cells**. *Sci Adv* 2020, **6**(19):eaaz0571.
51. Shapiro J, Iancu O, Jacobi AM, McNeill MS, Turk R, Rettig GR, Amit I, Tovim-Recht A, Yakhini Z, Behlke MA *et al*: **Increasing CRISPR Efficiency and Measuring Its Specificity in HSPCs Using a Clinically Relevant System**. *Mol Ther Methods Clin Dev* 2020, **17**:1097-1107.
52. Cerchione D, Loveluck K, Tillotson EL, Harbinski F, DaSilva J, Kelley CP, Keston-Smith E, Fernandez CA, Myer VE, Jayaram H *et al*: **SMOOT libraries and phage-induced directed evolution of Cas9 to engineer reduced off-target activity**. *PloS one* 2020, **15**(4):e0231716.
53. Hu Z, Wang S, Zhang C, Gao N, Li M, Wang D, Wang D, Liu D, Liu H, Ong SG *et al*: **A compact Cas9 ortholog from Staphylococcus Auricularis (SauriCas9) expands the DNA targeting scope**. *PLoS Biol* 2020, **18**(3):e3000686.
54. Walton RT, Christie KA, Whittaker MN, Kleinstiver BP: **Unconstrained genome targeting with near-PAMless engineered CRISPR-Cas9 variants**. *Science* 2020, **368**(6488):290-296.
55. Chen P, Zhou J, Wan Y, Liu H, Li Y, Liu Z, Wang H, Lei J, Zhao K, Zhang Y *et al*: **A Cas12a ortholog with stringent PAM recognition followed by low off-target editing rates for genome editing**. *Genome biology* 2020, **21**(1):78.
56. Kulcsar PI, Talas A, Toth E, Nyeste A, Ligeti Z, Welker Z, Welker E: **Blackjack mutations improve the on-target activities of increased fidelity variants of SpCas9 with 5'G-extended sgRNAs**. *Nat Commun* 2020, **11**(1):1223.
57. Toth E, Varga E, Kulcsar PI, Kocsis-Jutka V, Krausz SL, Nyeste A, Welker Z, Huszar K, Ligeti Z, Talas A *et al*: **Improved LbCas12a variants with altered PAM specificities further broaden the genome targeting range of Cas12a nucleases**. *Nucleic acids research* 2020, **48**(7):3722-3733.
58. Miller SM, Wang T, Randolph PB, Arbab M, Shen MW, Huang TP, Matuszek Z, Newby GA, Rees HA, Liu DR: **Continuous evolution of SpCas9 variants compatible with non-G PAMs**. *Nature biotechnology* 2020, **38**(4):471-481.
59. Song CQ, Jiang T, Richter M, Rhym LH, Koblan LW, Zafra MP, Schatoff EM, Doman JL, Cao Y, Dow LE *et*

- al: Adenine base editing in an adult mouse model of tyrosinaemia. Nat Biomed Eng* 2020, **4**(1):125-130.
60. Rose JC, Popp NA, Richardson CD, Stephany JJ, Mathieu J, Wei CT, Corn JE, Maly DJ, Fowler DM: **Suppression of unwanted CRISPR-Cas9 editing by co-administration of catalytically inactivating truncated guide RNAs.** *Nat Commun* 2020, **11**(1):2697.
61. Ou L, Przybilla MJ, Ahlat O, Kim S, Overn P, Jarnes J, O'Sullivan MG, Whitley CB: **A Highly Efficacious PS Gene Editing System Corrects Metabolic and Neurological Complications of Mucopolysaccharidosis Type I.** *Molecular therapy : the journal of the American Society of Gene Therapy* 2020, **28**(6):1442-1454.
62. Goldberg GW, Spencer JM, Giganti DO, Camellato BR, Agmon N, Ichikawa DM, Boeke JD, Noyes MB: **Engineered dual selection for directed evolution of SpCas9 PAM specificity.** *Nat Commun* 2021, **12**(1):349.
63. Yant SR, Wu X, Huang Y, Garrison B, Burgess SM, Kay MA: **High-resolution genome-wide mapping of transposon integration in mammals.** *Mol Cell Biol* 2005, **25**(6):2085-2094.
64. Turchiano G, Latella MC, Gogol-Doring A, Cattoglio C, Mavilio F, Izsvak Z, Ivics Z, Recchia A: **Genomic analysis of Sleeping Beauty transposon integration in human somatic cells.** *PloS one* 2014, **9**(11):e112712.
65. Moldt B, Miskey C, Staunstrup NH, Gogol-Doring A, Bak RO, Sharma N, Mates L, Izsvak Z, Chen W, Ivics Z *et al*: **Comparative genomic integration profiling of Sleeping Beauty transposons mobilized with high efficacy from integrase-defective lentiviral vectors in primary human cells.** *Molecular therapy : the journal of the American Society of Gene Therapy* 2011, **19**(8):1499-1510.
66. Kovac A, Miskey C, Menzel M, Grueso E, Gogol-Doring A, Ivics Z: **RNA-guided retargeting of Sleeping Beauty transposition in human cells.** *Elife* 2020, **9**.
67. Oh YK, Suh D, Kim JM, Choi HG, Shin K, Ko JJ: **Polyethylenimine-mediated cellular uptake, nucleus trafficking and expression of cytokine plasmid DNA.** *Gene Ther* 2002, **9**(23):1627-1632.

REVIEWERS' COMMENTS:

Reviewer #1 (Remarks to the Author):

The authors provided adequate answers to all of my concerns.
I recommend that this paper be published in communications biology.

Reviewer #2 (Remarks to the Author):

The authors of "Tag-seq: a convenient and scalable method for genome-wide specificity assessment of CRISPR/Cas nucleases" have done an admirable job at addressing responses to their initial manuscript. They express that there are difficulties in addressing all the concerns but were able to investigate and modify their study in significant ways. I have a few remaining comments / questions:

1. The comparison of operational costs for the different protocols is helpful in displaying differences, and the list of studies provided which demonstrate the field's bias toward MiSeq instruments is thorough. MiSeq / NextSeq instruments cost less overall to purchase and less to operate on a per run basis, though the output is much less than their counterpart systems, HiSeq / NovaSeq. For an academic lab, the lower entry cost is often preferable as the need for high through-put is not required and funding is likely limiting. This could explain the observed study bias, but amplification-based assays (such as GUIDE-seq) tend to "bottleneck" during their amplification. This leads to overall lower diversity in the sequencing library, or fewer unique templates. Another reason for using the MiSeq/NextSeq instruments is because often deeper sequencing than a million reads or so per sample is not necessary to identify off-target sites. Therefore, it can be difficult to justify "deeper" sequencing of GUIDE-seq like libraries as increased depth after a certain point yields little new information. The authors may have observed this in their work, specifically on Supplementary Figure 3 A and B. New sites identified at low frequencies cannot necessarily be verified, even by deep sequencing, and there is a level of stochasticity of sampling, in which a repeated sample may not yield all the same low-frequency observed sites. For a user to utilize this information, they need to figure out the optimal sequencing depth for their libraries, or an optimal range. That way they would be informed as to how many samples they may pool into a multiplex prior to sequencing to maximize their data output per cost of operation. Using data from Supplementary Figure 2, could the authors provide a rarefaction analysis per sample to support that moving GUIDE-seq like libraries to the HiSeq / NovaSeq instruments is even cost effective, or to support how many samples should be run to make it cost effective on those instruments?

2. The results from Figure 2 compare the Tag-seq read coverage at the different off-target sites measured from using either the Lonza or PEI based methods with several guide-RNAs. One observation here is that the proportions of reads at the different off-target sites are not the same between the two methods. Does this mean that off-targets may be observed at different frequencies based on the different methods? Does the different transfection method influence observation of off-targets? In each case (A, B, and C) there are more off-targets observed for PEI that were not observed with Lonza. Can this be interpreted as the transfection method may decrease the specificity of the nuclease? If the read count information is not useful for quantitative analysis, then how does this impact the questions raised in comment 1 above? Comments around this topic would be helpful in the discussion of the manuscript.

Responses to Reviewer

Reviewer #2 (Remarks to the Author):

The authors of “Tag-seq: a convenient and scalable method for genome-wide specificity assessment of CRISPR/Cas nucleases” have done an admirable job at addressing responses to their initial manuscript. They express that there are difficulties in addressing all the concerns but were able to investigate and modify their study in significant ways. I have a few remaining comments / questions:

1. The comparison of operational costs for the different protocols is helpful in displaying differences, and the list of studies provided which demonstrate the field’s bias toward MiSeq instruments is thorough. MiSeq / NextSeq instruments cost less overall to purchase and less to operate on a per run basis, though the output is much less than their counterpart systems, HiSeq / NovaSeq. For an academic lab, the lower entry cost is often preferable as the need for high through-put is not required and funding is likely limiting. This could explain the observed study bias, but amplification-based assays (such as GUIDE-seq) tend to “bottleneck” during their amplification. This leads to overall lower diversity in the sequencing library, or fewer unique templates. Another reason for using the MiSeq/NextSeq instruments is because often deeper sequencing than a million reads or so per sample is not necessary to identify off-target sites. Therefore, it can be difficult to justify “deeper” sequencing of GUIDE-seq like libraries as increased depth after a certain point yields little new information. The authors may have observed this in their work, specifically on Supplementary Figure 3 A and B. New sites identified at low frequencies cannot necessarily be verified, even by deep sequencing, and there is a level of stochasticity of sampling, in which a repeated sample may not yield all the same low-frequency observed sites. For a user to utilize this information, they need to figure out the optimal sequencing depth for their libraries, or an optimal range. That way they would be informed as to how many samples they may pool into a multiplex prior to sequencing to maximize their data output per cost of operation. Using data from Supplementary Figure 2, could the authors provide a rarefaction analysis per sample to support that moving GUIDE-seq like libraries to the HiSeq / NovaSeq instruments is even cost effective, or to support how many samples should be run to make it cost effective on those instruments?

We thank the reviewer for the constructive and helpful comments. We agree with that MiSeq/NextSeq instruments cost less overall to purchase and less to operate on a per run basis, albeit with a lower through-put, it is a suitable choice

for the academic labs who intent to own themselves sequencing machines because MiSeq/NextSeq instruments almost meet the need of the GUIDE-seq libraries sequencing and generates sufficient reads for off-target sites identification. Additionally, for the question of the sequencing depth, in our experiments, we usually sequence a library with a depth of 2-3 G raw data in Hiseq/Novaseq. Because this depth lead to the comparable counts detected by the GUIDE-seq (Supplementary Figure 2). Thus, we recommend this depth (2-3 G raw data) for Tag-seq sequencing using the HiSeq/NovaSeq instruments. However, if you want to obtain more counts, you can sequence it much deeper. Finally, the question about the pooling sample before sequencing is not as a necessary procedure of our Tag-seq, because our method is perfectly fit the sequencing market niche and we usually commercially sequence the libraries, which can flexibly combine with other customers' samples with various libraries, from a single one to more than hundreds. Therefore, the Tag-seq will be more convenient and cost-effective than these who own themselves sequencing device, such as Miseq or NextSeq.

2. The results from Figure 2 compare the Tag-seq read coverage at the different off-target sites measured from using either the Lonza or PEI based methods with several guide-RNAs. One observation here is that the proportions of reads at the different off-target sites are not the same between the two methods. Does this mean that off-targets may be observed at different frequencies based on the different methods? Does the different transfection method influence of observation of off-targets? In each case (A, B, and C) there are more off-targets observed for PEI that were not observed with Lonza. Can this be interpreted as the transfection method may decrease the specificity of the nuclease? If the read count information is not useful for quantitative analysis, then how does this impact the questions raised in comment 1 above? Comments around this topic would be helpful in the discussion of the manuscript.

Thanks for the reviewer's comments. As a general rule, the longer time of both Cas9 and sgRNA are stayed in cells, the more opportunity of off-target effects will obtain. Thus, Off-target effect of CRISPR tools is observed at various extent when delivered by different methods, such as plasmid transfection, virus-based transduction or ribonucleoproteins (RNP) delivery. Use of the RNP complex delivery system, which can efficiently mediate genome editing in a manner of "fast on, fast off", has been demonstrated as an effective method for improving the cleavage specificity¹⁻³. Similarly, the Lonza electroporation is a nucleofector, which can mediate plasmids into the nuclei and lead to efficient expression; While the PEI method is a common transfection required a long time of process

including cellular uptake, nuclear trafficking and subcellular retention⁴. Additionally, previous works about gene editing in stem cells has been also proved that electroporation is more efficient than other transfection methods (such as liposomes transfection)⁵. Thus, it is reasonable for the differences observed in these two methods. We have added these explanation in the result section in the revised manuscript.

1. Kim, S., Kim, D., Cho, S.W., Kim, J. & Kim, J.S. Highly efficient RNA-guided genome editing in human cells via delivery of purified Cas9 ribonucleoproteins. *Genome Res* **24**, 1012-1019 (2014).
2. Liang, Z. et al. Efficient DNA-free genome editing of bread wheat using CRISPR/Cas9 ribonucleoprotein complexes. *Nat Commun* **8**, 14261 (2017).
3. Vakulskas, C.A. & Behlke, M.A. Evaluation and Reduction of CRISPR Off-Target Cleavage Events. *Nucleic Acid Ther* **29**, 167-174 (2019).
4. Oh, Y.K. et al. Polyethylenimine-mediated cellular uptake, nucleus trafficking and expression of cytokine plasmid DNA. *Gene Ther* **9**, 1627-1632 (2002).
5. Liang, X. et al. Rapid and highly efficient mammalian cell engineering via Cas9 protein transfection. *J Biotechnol* **208**, 44-53 (2015).